# Unified Medical Image Pre-training in Language-Guided Common Semantic Space

## Abstract

Vision-Language Pre-training (VLP) has shown the merits of analysing medical images, by leveraging the semantic congruence between medical images and their corresponding reports. It efficiently learns visual representations, which in turn facilitates enhanced analysis and interpretation of intricate imaging data. However, such observation is predominantly justified on single-modality data (mostly 2D images like X-rays), adapting VLP to learning unified representations for medical images in real scenario remains an open challenge. This arises from medical images often encompass a variety of modalities, especially modalities with different various number of dimensions (e.g., 3D images like Computed Tomography). To overcome the aforementioned challenges, we propose an **U**nified **Med**ical **I**mage Pre-training framework, namely *UniMedI*, which utilizes diagnostic reports as common semantic space to create unified representations for diverse modalities of medical images (especially for 2D and 3D images). Under the text's guidance, we effectively uncover visual modality information, identifying the affected areas in 2D X-rays and slices containing lesion in sophisticated 3D CT scans, ultimately enhancing the consistency across various medical imaging modalities. To demonstrate the effectiveness and versatility of *UniMedI*, we evaluate its performance on both 2D and 3D images across 10 different datasets, covering a wide range of medical image tasks such as classification, segmentation, and retrieval. *UniMedI* has demonstrated superior performance in downstream tasks, showcasing its effectiveness in establishing a universal medical visual representation.

## 1 Introduction

In recent years, the field of medical image analysis has witnessed significant advancements, largely driven by the application of deep learning techniques and the increasing availability of medical imaging data. Notably, Visual-Language Pre-training (VLP) (Huang et al., 2021; Boecking et al., 2022; Bannur et al., 2023) attracts lots of attention, as it reduces the need for costly and time-consuming manual annotations by leveraging the vast amount of information in radiology reports and unlabeled data. Despite these success, further expanding the data scale for medical VLP remains non-trivial, because the availability of single-modality medical images is limited, especially when compared to the general domain. This introduces a strong need to integrate multi-modality medical images (e.g., X-rays, Computed Tomography (CT) and Magnetic Resonance Imaging(MRI)) within a unified VL framework. However, fully leveraging the information across multi-modal images within this VL framework is unexplored.

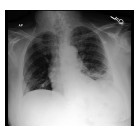

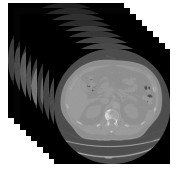

Lower lung atelectasis with probable left lower lobe pneumonia. Mild edema difficult to exclude. PA and lateral views of the chest provided. Airspace consolidation in the left lower lung is concerning for pneumonia likely within the left lower lobe…

Chest angiogram was performed according to the pulmonary thromboembolism protocol and showed normal pulmonary arteries with no filling defects. Dull glass lesion and consolidative area in the right upper lobe suggested pneumonia...

Figure 1: An example showing X-ray (up) and CT scan (down) both demonstrate similar abnormality, recording in the report.

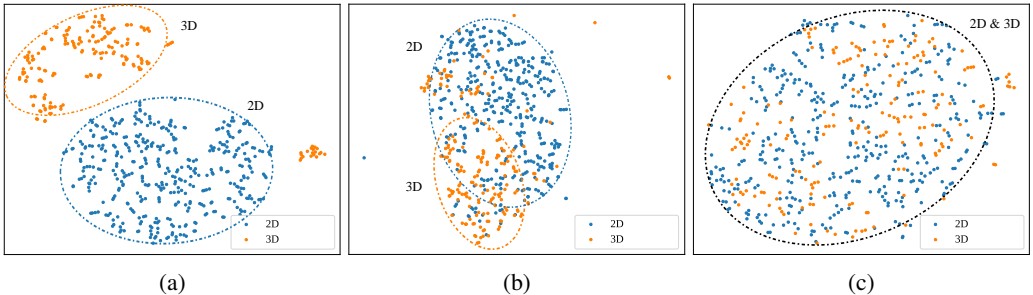

Figure 2: t-SNE visualizations of image representations by models trained with different methods (2D: X-rays, 3D: CT, both modalities denote the same disease, pneumonia.). (a) Two models for different image modalities are trained individually in separate VLP process. (b) One models for different image modalities are trained in one VLP processes, but without designes in *UniMedI*. (c) *UniMedI*. Learning a common semantic space for different medical images is non-trivial, even with language guidance, and *UniMedI* can well handle this integration. We use circles to highlight differences between different images.

On the above aspect, the inherent heterogeneity of medical imaging from different modalities obstructs their effective integration. One obvious and important problem is that medical images have different dimensions. For example, X-rays are 2D images, while CT scans are 3D images. To tackle this challenge, we start from the following key observation: *despite big differences, medical images from various modalities share a common semantic latent space, which captures the underlying features of an individual's health status, and such status are reflected in medical reports via language.* As shown in Fig. 1, the X-ray and CT scan can contribute to a comprehensive understanding of pneumonia, reflecting the commonality within the latent space, and these abnormalities are listed in reports. This observation motivate us to *map data from various medical image modalities into the shared semantic space, which is guided by language in reports.* This strategy not only tackles data-related issues but also fosters synergy and collaboration among distinct modalities, ultimately resulting in a more holistic understanding of an individual's health condition.

However, creating a unified model that effectively maps data from different sources into a common space for combined learning is challenging, even with language guidance in reports. Figure 2a demonstrates the representation space of two distinct modalities with different dimensions (i.e., 2D X-rays and 3D CT scans) when trained individually via VLP. They are far apart in the representation space, even with same pathological information in reports. Furthermore, Figure 2b shows simply unifying them in one model does not solve the problem. Although the distance between representations of two modalities are shortened to some extent, their representations remain insufficiently compact, since only little space are shared between them.

To address the above challenge, we propose *UniMedI*, a novel **Uni**fied VL framework, designed to effectively integrate **Med**ical multi-modal **I**mages into a language-guided common semantic space. First, under the dilemma that paired 2D and 3D medical images are unavailable, and naively integration is not effectively as we shown above, we first design an attentive selection method to accurately identify text-relevant 2D slices without extra annotations. This builds a data bridge between 2D and 3D medical images. Then, we devise a cross-dimensional VLP method to bring both 3D data and selected 2D slices closer to the same report representation space, constructing a unified VL framework. Moreover, we introduce a self-distillation technique using a teacher-student structure and construct a masking and recovery task, further enhancing the associations between 2D and 3D data within the image space. Figure 2c shows *UniMedI* significantly reduces the distance between 2D and 3D features after undergoing our effective design for cross-dimensional pre-training.

To further demonstrate the effectiveness of our approach, we conduct extensive visualizations and experiments to showcase the working mechanisms and superior representational capabilities of our model. We evaluate our *UniMedI* framework on 10 real-world medical datasets and various downstream tasks (i.e., classification, segmentation and retrieval). The results consistently show superior performance, regardless of whether *UniMedI* is applied to full-scale data or limited data scenarios. We also provide visualizations on regions and slices selected by *UniMedI*, verifying our claim that *UniMedI* can identify key information from both 2D and 3D medical images.

## 2 RELATED WORK

**Medical Self-supervised Learning**   In the domain of medical image analysis, a number of self-supervised learning (SSL) techniques have been developed to exploit the unique characteristics of medical data. These methods construct feature embedding spaces by designing pre-text tasks, such as solving jigsaw puzzles Noroozi & Favaro and inpainting tasks Pathak et al. (2016). Recently, researchers have explored the use of 3D convolutional neural network (CNN) architectures while retaining established SSL tasks on 2D CNNs Tang et al. (2022). However, the diversity of medical data poses a significant challenge, as the development of a unified visual representation that adequately captures the intricacies of different data types remains a crucial yet complex task that requires further investigation. To address this challenge, Xie et al. (2022) proposed Unimiss, a universal medical self-supervised representation learning framework that overcomes the dimensionality barrier. Furthermore, Nguyen et al. (2023) introduced Joint, an SSL framework capable of accommodating various data dimensions and generating versatile pre-trained weights for both 2D and 3D downstream applications. These approaches have made notable contributions to handling data from different modalities. However, they have given relatively less attention to the relationships and connections between different types of medical data.

**Medical Vision-Language Processing**   Medical Vision-Language Processing (VLP) has emerged as a promising approach for learning medical visual representations by leveraging naturally occurring paired descriptive text Zhang et al. (2022). Huang et al. (2021) propose Gloria, an attention-based framework that contrasts image sub-regions and words in the paired report to learn global and local representations. Wang et al. (2022) further optimize the framework from the perspective of disease in their method MGCA. These methods exhibit remarkable performance in various downstream tasks involving medical images. However, the application of medical VLP is primarily limited to 2D images, mainly due to the limited availability of extensive 3D medical image-text datasets. Compared to 2D medical image-text pairs, 3D images and reports contain more abundant information, which offers clear advantages for learning visual representations. While some methods Liu et al. (2023); Chen et al. (2023) attempt to address this limitation by converting 3D data into 2D slices and subsequently employing generative models to generate captions for 3D medical data, this approach results in a loss of the original 3D volume structure information. Therefore, it is imperative to develop strategies that can effectively harness the valuable information present in 3D images and reports while preserving the structural integrity of the data. This will facilitate the enhancement of the learning process for visual representations in medical VLP.

## 3 METHODOLOGY

Figure 3 illustrates *UniMedI* and its designs to realize integration of 2D and 3D medical images. Generally, to overcome the challenges that no paired 2D and 3D image data exists, *UniMedI* employes the following pipeline. When the input is a 3D volume, we first extract a portion of 2D slices from it which most relevant to the report, and then regard the selected slices as 2D image. Those selected 2D slices are fed into the network along with the original 3D volume, allowing us to jointly learn the relationships between 2D, 3D, and radiology reports, and ultimately form a unified feature space. When the input is a 2D image, the slice selection process is omitted.

In Section 3.1, we demonstrate our designed attentive slice selection method, which can identify more relevant 2D slices in 3D data related to the report text, helping us learn the unified space between 2D and 3D data guided by report. In Section 3.2, we design a method to bring together 3D data and selected 2D slices closer to the same report representation, which serves as the foundation for our language-guided construction of a unified model. In Section 3.3, we design a self-distillation technique to EMA teacher for the visual encoder, constructing image-level and patch-level contrastive learning tasks, further enhancing the connection between 2D and 3D data.

### 3.1 ATTENTIVE SLICE SELECTION

In order to construct a cross-modal unified representation space, we have chosen language as the bridge. Therefore, we need to extract key information from various image modalities that correspond to the information in medical reports. Particularly, important 2D slices relevant with reports should

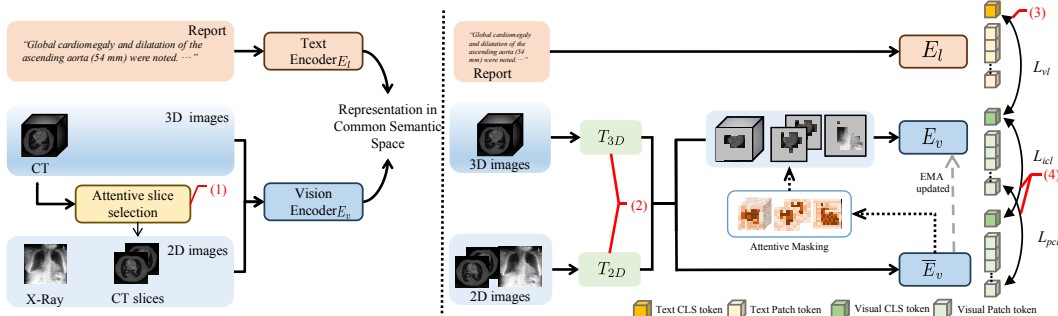

Figure 3: Illustration of the proposed *UniMedI* framework. To effectively integrate multi-modal medical images in a language-guided common semantic space, *UniMedI* incorporates following designs. (1) An strategy of attentive slice selecting from 3D volume to bridge 2D and 3D images even without paired 2D and 3D data (details shown in Fig. 4). The concatenated inputs of 2D and 3D allow us to perform joint modeling across dimensions. (2) Shared backbone $E_v$ for 2D and 3D images and separate tokenizer $T_{2D}$ and $T_{3D}$. (3) Language-guidance for unified image representation provided by the language encoder $E_l$ and vision-language loss $L_{vl}$. (4) Self-distillation (implemented by image contrastive learning loss $L_{icl}$ and patch contrastive learning loss $L_{pcl}$) to enhance interactions between images tokens from different modalities. The distillation target comes the teacher network $\overline{E}_v$, which is updated by exponential moving averaged (EMA) over the student network $E_v$.

be selected from 3D volume. This process is similar to how doctors view CT scans; they also base their report descriptions on some important slices.

As shown in Figure 4, in order to better locate the lesion-related 2D slices in the 3D data, we use the attention weights of the $[CLS]$ token in the EMA teacher as the basis for calculation. The visual encoder's $[CLS]$ token is directly supervised by the radiology report features from the language encoder, reflecting the most likely lesion areas described in the report. For the attentive score at token location $P$:

$$s^P = \frac{1}{HL} \sum_{l=1}^{L} \sum_{h=1}^{H} \text{Softmax}\left( \frac{\mathbf{f}_{lh}^q(CLS) \cdot \mathbf{f}_{lh}^k(P)}{\sqrt{C}} \right), \tag{1}$$

where $l$ denotes the layer index; $h$ denotes the attention head index; $\mathbf{f}_{lh}^q(CLS)$ denotes the query embedding of the $[CLS]$ token at Layer $l$ and Head $h$; $\mathbf{f}_{lh}^k(P)$ denotes the key embedding of Layer $l$ and Head $h$ for an 3D image token at location $P$; $C$ is the number of channels for the query and key embedding.

The important slices located strategy is based on the token-level score. Each token in the original CT volume represents a small voxel. By aggregating scores based on the slice dimension, we can calculate the total score for each group of slices:

$$s_i = \frac{1}{N} \sum_{j=1}^{N} s_{ij}^P, \tag{2}$$

where $s_i$ is the attentive score for the $i$-th slice, $s_{ij}^P$ is the token-level attentive score for the $j$-th voxel in $i$-th slice, $N$ represents the total number of voxels included in a slice. After aggregating the attentive scores, we can obtain text relevance scores for each 2D slice. We then choose the top $k$ slices to establish a connection with the 3D data and the report, allowing us to learn a shared feature space.

## 3.2 CROSS-DIMENSIONAL MEDICAL VISUAL-LANGUAGE PRETRAINING.

We use CLIP Radford et al. (2021) loss for cross-modal pre-training of 2D and 3D medical images and their corresponding reports. CLIP is a powerful tool that enables the alignment of features from two modalities after large-scale contrastive learning. For 2D X-ray training, we directly use $T_{2D}$ and $E_v$ for feature extraction, obtaining the global image feature $[CLS]$ token, and then aligning it with the language encoder $E_l$'s $[CLS]$ token. For the training of 3D CT scan data, the 2D slices within it also carry the content of the same radiology report, so we select attentive 2D slices according to

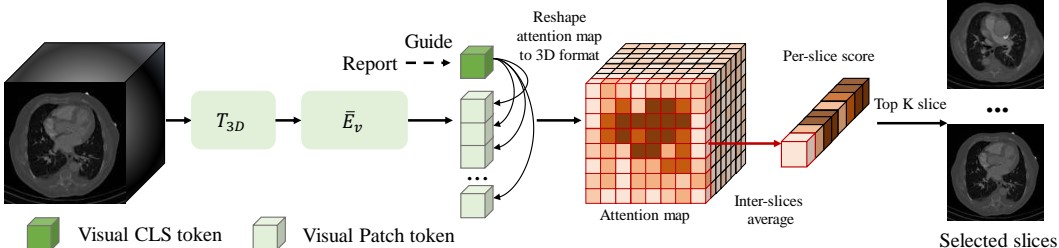

Figure 4: Attentive slice selection from 3D volume. Generally, the slice is selected according to the attention weights of the [CLS] token attending to other tokens, and the [CLS] token is also guided by language in the report. We compute the average attention weights within each sliced area, and then select the top K slices with the highest scores.

the method in Section 3.1 as joint input. Through this approach, we bring the 2D slice features and 3D features closer to the same language encoder's features, using radiology reports as a medium to form cross-dimensional interactions.

A highlight of our work is the use of attentive slices selection to ensure that the selected 2D slices are sufficiently representative. Only in this way can these 2D slices carry the supervision information from the report and, together with the 3D features, construct a joint feature space. If we were to use random selection, it would be easy to cause mismatches between the visual and textual information, and the noise generated would make the model's understanding on 2D data very confusing. Once the common coordinates from the report are no longer accurate, it would not be possible to effectively form a cross-dimensional information bridge.

### 3.3 ENHANCING DIMENSIONAL INTERACTIONS VIA SELF-DISTILLATION

In Section 3.1, we introduced the method for selecting 2D slices that can share the same report. Then, in Section 3.2, we aligned them across dimensions using text as shared coordinates for visual-textual training. In fact, apart from using text as a medium, the projected representative 2D slice features and 3D features with global information also possess strong correlations. We aim to construct an auxiliary task to directly leverage this correlation, further enhancing the cross-dimensional communication.

We adopted a simple and straightforward auxiliary task design: mask and recovery. We chose to use the self-distillation method for implementation Yang et al. (2023); Zhou et al. (2021), due to its simplicity and effectiveness. During the training process, we mask a certain proportion of both 2D and 3D tokens in the online encoder, while keeping the complete input in the EMA encoder. Therefore, this non-trivial task requires predicting the EMA encoder's features directly from the online encoder, as there is a significant amount of missing information. For both 2D and 3D recovery tasks, the model has to learn the correlation with the other modality to obtain more reference information, thus directly strengthening the interaction between 2D and 3D features within the network.

Similarly, during the token masking phase, we also employed the attentive selection design. While passing through the EMA encoder, we calculated the patch scores as described in Equation 1, and retained the portion with the highest scores. This approach minimizes the disruption of effective lesion structures, thereby avoiding ambiguity and making the cross-modal interaction more meaningful.

During the feature distillation process, we utilized the head and loss from BYOL Grill et al. (2020). We applied this loss to both the global [CLS] tokens and all local patch tokens in the output 2D and 3D features, thereby enabling interaction at different granularities to enhance feature robustness.

## 4 EXPERIMENTS

We build our universal medical framework *UniMedI* and pre-train on the two medical vision-report datasets with different modalities including 2D X-rays and 3D CT scans. Furthermore, extensive experiments on multiple cross-modal downstream dataset from diverse tasks are conducted to verify

| Method | CheXpert(AUC) | | | RSNA(AUC) | | | COVIDx(ACC) | | |
|---|---|---|---|---|---|---|---|---|---|
| | 1% | 10% | 100% | 1% | 10% | 100% | 1% | 10% | 100% |
| Random Init | 56.1 | 62.6 | 65.7 | 58.9 | 69.4 | 74.1 | 50.5 | 60.3 | 70.0 |
| ImageNet Init | 74.4 | 79.9 | 81.4 | 74.9 | 74.5 | 76.3 | 64.8 | 78.8 | 86.3 |
| *pre-trained on CheXpert* | | | | | | | | | |
| DSVE Engilberge et al. (2018) | 50.1 | 51.0 | 51.5 | 49.7 | 52.1 | 57.8 | - | - | - |
| VSE++ Faghri et al. (2017) | 50.3 | 51.2 | 52.4 | 49.4 | 57.2 | 57.9 | - | - | - |
| GLoRIA Huang et al. (2021) | 86.6 | 87.8 | 88.1 | 86.1 | 88.0 | 88.6 | 67.3 | 77.8 | 89.0 |
| *pre-trained on MIMIC-CXR* | | | | | | | | | |
| Caption-Transformer Cornia et al. (2020) | 77.2 | 82.6 | 83.9 | - | - | - | - | - | - |
| Caption-LSTM Xu et al. (2015) | 85.2 | 85.3 | 86.2 | - | - | - | - | - | - |
| Contrastive-Binary Tan & Bansal (2019) | 84.5 | 85.6 | 85.8 | - | - | - | - | - | - |
| ConVIRT Zhang et al. (2022) | 85.9 | 86.8 | 87.3 | 77.4 | 80.1 | 81.3 | 72.5 | 82.5 | 92.0 |
| GLoRIA-MIMIC Huang et al. (2021) | 87.1 | 88.7 | 88.0 | 87.0 | 89.4 | 90.2 | 66.5 | 80.5 | 88.8 |
| MGCA (ResNet-50) Wang et al. (2022) | 87.6 | 88.0 | 88.2 | 88.6 | 89.1 | 89.9 | 72.0 | 83.5 | 90.5 |
| MGCA (ViT-B/16) Wang et al. (2022) | 88.8 | 89.1 | 89.7 | 89.1 | 89.9 | 90.8 | 74.8 | 84.8 | 92.3 |
| ***UniMedI* (Ours, ViT-B/16)** | **89.4** | **89.7** | **90.5** | **90.0** | **90.4** | **91.5** | **80.3** | **92.4** | **94.6** |

Table 1: Linear classification results on CheXpert, RSNA and COVIDx with 1%, 10%, 100% training data. Area under ROC curve (AUROC [%]) are reported for CheXpert and RSNA dataset, and accuracy (ACC [%]) is reported for COVIDx dataset. The best results are highlighted in **boldface**.

the effectiveness of the multi-modal vision representations. In the following subsections, we first present the pre-training experiments settings in Section 4.1 and two main downstream tasks in Section 4.2. In addition, we compare the performance of our proposed approach with the state-of-the-art vision-language processing methods in Section 4.3. Finally, we perform plenty of ablation experiments on multi-modal downstream tasks and visualization to show the validity of each module of our framework.

## 4.1 PRE-TRAINING SETUP

**Dataset** We pre-train our *UniMedI* framework on the JPG version of 2D X-rays dataset **MIMIC-CXR 2.0.0** Johnson et al. (2019) and the MINC version of 3D CT scans dataste **BIMCV** de la Iglesia Vayá et al. (2021). As the downstream 2D datasets only encompass frontal-view chest images, we remove the lateral-view images to preprocess the 2D dataset **MIMIC-CXR 2.0.0**. Similarly, as the downstream 3D datasets only encompass frontal-view chest images, we remove the lateral-view images to preprocess the 3D dataset **BIMCV**. For the processing of text reports, we remove the reports which are less than 3 tokens for 2D and 3D datasets following Wang et al. (2022).

**Implementation Details** Following Gloria Huang et al. (2021), we utilize ViT-B/16 Dosovitskiy et al. (2020) as the vision encoder to extract representations in the common feature space for 2D and 3D visual data. We use BioClinicalBERT Alsentzer et al. (2019) as the text encoder to obtain the report embeddings.

| Method | CC-CCII | | |
|---|---|---|---|
| | 1% | 10% | 100% |
| Random Init | 43.4 | 69.7 | 74.8 |
| UniMiSS* Xie et al. (2022) | 41.6 | 73.1 | 84.1 |
| *UniMedI** | 64.2 | 75.1 | 84.9 |
| ***UniMedI*** | **75.6** | **84.8** | **89.4** |

Table 2: Linear classification results on CC-CCII with 1%, 10%, 100% training data. Accuracy are reported for the dataset. * denotes the input size $16 \times 96 \times 96$. Others is $32 \times 128 \times 128$. The best results are highlighted in **boldface**.

| Method | CC-CCII | LUNA |
|---|---|---|
| *supervised* | | |
| ResNet3D101 | 85.5 | - |
| CovidNet3D-L | 88.7 | - |
| *unsupervised* | | |
| Joint Nguyen et al. (2023) | - | 94.2 |
| ***UniMedI*** | **93.8** | **95.9** |

Table 3: Classification results on CC-CCII, RI-CORD with full training data. ACC [%] is reported for CC-CCII and AUC [%] is reported for LUNA2016-v2. The best results are highlighted in **boldface**.

Table 4: Ablation study of training mode on linear classification (2D dataset CheXpert, RSNA and 3D dataset CC-CCII) settings. We report Area under ROC curve (AUROC [%]) on CheXpert and RSNA datasets, and (Acc [%]) on CC-CCII dataset. Best results of each setting are in boldface.

| Training tasks | | CheXpert (AUC) | | | RSNA (AUC) | | | CC-CCII (Acc) | | |
|---|---|---|---|---|---|---|---|---|---|---|
| 2D | 3D | 1% | 10% | 100% | 1% | 10% | 100% | 1% | 10% | 100% |
| ✓ | | 87.1 | 88.0 | 88.4 | 88.7 | 89.5 | 90.3 | - | - | - |
| | ✓ | - | - | - | - | - | - | 55.6 | 71.7 | 76.4 |
| ✓ | ✓ | **87.4** | **88.1** | **88.5** | **88.9** | 89.3 | **90.6** | **72.4** | **80.0** | **86.2** |

## 4.2 DOWNSTREAM TASKS AND EXPERIMENTAL SETUP

**Medical Classification**    We conduct medical image classification on three representative datasets: (1) **CheXpert** Irvin et al. (2019), which contains 191,229 frontal-view chest radiographs. The task is to classify each image into 5 individual binary labels: *atelectasis*, *cardiomegaly*, *consolidation*, *edema*, and *pleural effusion*. Following Zhang et al. (2022); Huang et al. (2021), we hold out the expert-labeled validation set as test data and randomly select 5,000 radiographs from training data for validation. (2) **RSNA** Pneumonia Shih et al. (2019). We use the stage 2 version, which contains around 29,700 frontal view chest radiographs. The task is a binary classification, *i.e.*, classifying each chest image into *normal* or *pneumothorax positive*. Following Huang et al. (2021), we manually split the dataset into training, validation, and test set with 70%/15%/15% ratio. (3) **COVIDx** Wang et al. (2020), which contains over 30*k* CXR images from a multinational cohort of over 16,600 patients. This dataset contains 16,490 positive COVID-19 images from over 2,800 patients. We use the latest version 6 of this dataset. The task is a three-class classification, *i.e.*, classifying each radiograph into *COVID-19*, *non-COVID pneumonia* or *normal*. We use the original validation dataset as test data and manually split 10% of original training set for validation.

Table 5: Ablation study of our framework on linear classification (2D dataset CheXpert, RSNA and 3D dataset CC-CCII) settings. We report Area under ROC curve (AUROC [%]) on CheXpert and RSNA datasets, and (Acc [%]) on CC-CCII dataset. $VL$ represents the default experiment setting include image-text contrastive loss $L_{vl}$ with random slices selection. $FD$ will include $L_{icl}$ and $L_{pcl}$ loss to execute self feature distillation. $Attn$ will use attentive slices selection instead of random. Best results of each setting are in boldface.

| Training tasks | | | CheXpert (AUC) | | | RSNA (AUC) | | | CC-CCII (Acc) | | |
|---|---|---|---|---|---|---|---|---|---|---|---|
| $VL$ | $FD$ | $Attn$ | 1% | 10% | 100% | 1% | 10% | 100% | 1% | 10% | 100% |
| ✓ | | | 87.4 | 88.1 | 88.5 | 88.9 | 89.3 | 90.6 | 72.4 | 80.0 | 86.2 |
| ✓ | ✓ | | 89.0 | 89.3 | 90.1 | 89.5 | 90.1 | 91.2 | 74.6 | 80.9 | 86.7 |
| ✓ | ✓ | ✓ | **89.4** | **89.7** | **90.5** | **90.0** | **90.4** | **91.5** | **75.6** | **84.8** | **89.4** |

We conduct medical volume classification on two representative datasets: (1) **CC-CCII** Zhang et al. (2020) and **LUNA 16** Setio et al. (2017). More details about the 3D datasets are in Appendix.

we use the *Linear Classification* setting to evaluate the representative ability of our universal vision-language pre-training framework. Apart from this, we also apply *Classification* to evaluate *UniMedI* for 3D data. *Linear Classification* freezes the pre-trained ViT vision encoder and only training a randomly initialized linear classification head for the downstream classification task with 1%, 10%, and 100% training data on each classification dataset.

**Medical Semantic Segmentation**    We conduct experiments to evaluate the performance of our framework for medical semantic segmentation on RSNA and BCV datasets: (1) RSNA Pneumonia Shih et al. (2019), contains 29700 frontal view radiograph. The task is to predict bounding boxes indicating evidence of pneumonia. We randomly split the original training set into 16,010/5,337/5,337 for training/validation/testing. We convert object detection ground truths into masks for semantic segmentation. (2) BCV Landman et al. (2015), which consists of 50 CT scans and is divided into 24/26 for training/testing following Xie et al. (2022).

We evaluate the segmentation performance with the paradigm that we use the pre-trained vision encoder as a frozen encoder and train a decoder portion using 1%, 10% and 100% training data

on RSNA dataset and 20%, 40%, 100% training data on BCV dataset. Dice scores are reported to evaluate the segmentation performance.

## 4.3 RESULT

### 4.3.1 RESULTS ON MEDICAL CLASSIFICATION

**2D Medical Image Classification** Table 1 reports the results of *Linear Classification* on three 2D medical image classification datasets (CheXpert, RSNA and COVIDx). The results of other methods on CheXpert and RSNA are from original paper Wang et al. (2022). The methods including *UniMedI* shown in the table are pre-trained on MIMIC-CXR dataset, which achieves a fair comparison. As for the state-of-the-art method, MGCA, we mainly compare the performance with the MGCA (ViT-B/16) which employs the ViT as the visual encoder. It is obvious that our method shows the best performance in the three 2D medical image classification for the different training data ratio (1%, 10%, 100%), outperforming the state-of-the-art MGCA (ViT-B/16) by a large margin. Specifically, our method outperforms MGCA with ViT-B/16 backbone with +0.6%, +0.6%, +0.8% AUROC on CheXpert dataset, +0.9%, +0.5%, +0.7% AUROC on RSNA dataset and +5.5%, +7.6%, +2.3% ACC on COVIDx dataset under the 1%, 10%, 100% training ratio respectively. The significant improvement indicates the data efficiency and effectiveness of our method.

**3D Medical Volume Classification** Table 2 reports the results of *Linear Classification* on the 2D medical image classification dataset, CC-CCII. We compare *UniMedI* with UniMiss Xie et al. (2022). To our knowledge, the UniMiSS Xie et al. (2022) is the state-of-the-art unified method to process 2D and 3D medical images. We show the performances of both UniMiSS and *UniMedI*, where the results are that our method achieves a +22.6%, +2.0% and +0.8% ACC gain on CC-CCII dataset comparing with the UniMiSS under the 1%, 10%, 100% training ratio respectively. The significant improvement indicates the data efficiency and effectiveness of our method.

When fine-tuning the total vision encoder and the linear classification head with full training data, as listed in Table 3, our method gets the best performance on the multiple 3D medical volume classification datasets (CC-CCII and LUNA2016-v2) compared with other methods. It is observed that our method achieves with 93.8% ACC on CC-CCII dataset, and 95.9% ACC on LUNA2016-v2 dataset respectively. The remarkable performance of our method shows the generalization of our method for 2D and 3D medical classification tasks. It demonstrates our framework possesses the ability of extracting universal features for multi-modal data.

| | RSNA | | |
|---|---|---|---|
| Method | 1% | 10% | 100% |
| ConVIRT | 55.0 | 67.4 | 67.5 |
| GLoRIA | 59.3 | 67.5 | 67.8 |
| GLoRIA-MIMIC | 60.3 | 68.7 | 68.3 |
| MGCA | 88.6 | 81.2 | 94.3 |
| MGCA (ViT-B/16) | 66.2 | 71.3 | 73.6 |
| *UniMedI* (ViT-B/16) | **67.8** | **73.1** | **75.3** |

Table 6: 2D Semantic segmentation results (Dice [%]) on RSNA with 1%, 10% and 100% training labels. Best results of each setting are in boldface.

| | BCV | | |
|---|---|---|---|
| Method | 20% | 40% | 100% |
| MoCo v3 | 74.5 | 78.2 | 82.0 |
| DINO | 75.3 | 78.9 | 82.6 |
| UniMiSS | **78.0** | 81.0 | 85.0 |
| *UniMedI* | 77.5 | **81.6** | **85.4** |

Table 7: 3D Semantic segmentation results (Dice [%]) on BCV with 20%, 40% and 100% training labels. Best results of each setting are in boldface.

### 4.3.2 RESULTS ON MEDICAL SEMANTIC SEGMENTATION

Table 6 and Table 7 report the results of *Semantic Segmentaion* on 2D and 3D medical data. In 2D semantic segmentation task, our method *UniMedI* significantly outperforms the current state-of-the-art algorithm, MGCA. When using 1% training data, *UniMedI* achieves 67.8% Dice, surpasssing the MGCA 1.6%. Meanwhile, concurrently, *UniMedI* also demonstrates exceptional performance in 3D semantic segmentation tasks. In the BCV dataset, *UniMedI* achieves 0.6% and 0.4% performance gain under 20% and 40% label settings compared with Unimiss. These results underscore the exceptional performance of our method in dense prediction tasks.

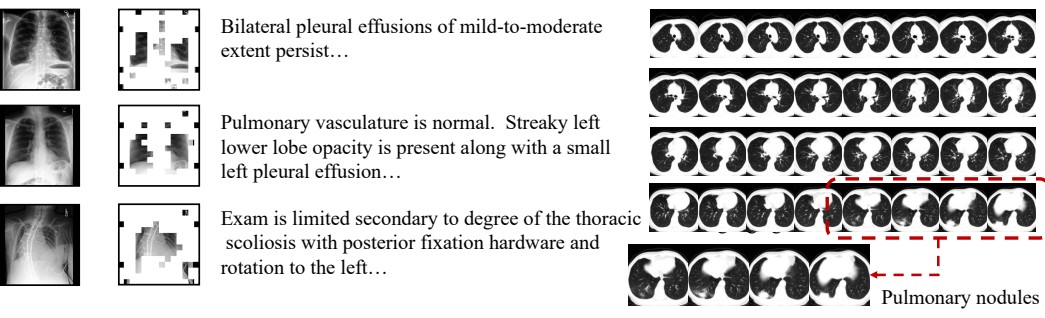

Figure 5: Visualization of mask and slices selection result under the guidance of language.

## 4.4 ANALYSIS OF OUR FRAMEWORK

**Visualization** To better demonstrate the effectiveness of our selection process guided by language, we visualize the original X-rays, masked X-rays, their corresponding reports, and original CT scans, as well as the selected lesion slices in Figure 5. On the left side of Figure 5, the first row effectively demonstrates how *UniMedI* accurately captures the areas referenced in the report, including the "Normal post-operative alignment of the sternal wires" and "Bilateral pleural effusions of mild-to-moderate extent persist". In addition, the second and third cases adeptly showcase the detection of pleural effusion and scoliosis, further emphasizing the method's precision. The right side of Figure 5 displays the comprehensive slice selection process employed by *UniMedI*. Amidst the extensive collection of CT scan slices, our method exhibits remarkable accuracy in pinpointing the slices containing lesions. As an example, the presence of pulmonary nodules is clearly noticeable in slices 28-31.

**Ablation Study of Component Design** We conduct ablation experiments primarily focusing on two aspects: training mode and framework module.

**Training mode** We pre-train our framework separately using only 2D data, only 3D data, and a combination of 2D and 3D data. Subsequently, we evaluated the performance on downstream 2D dataset CheXpert, RSNA and 3D dataset CC-CCII on linear classification task respectively, with the results presented in Table 3. It can be observed that the pretraining approach combining 2D and 3D data yields benefits for both single-modal 2D and 3D data classification tasks. Particularly, the enhancement achieved with the use of multimodal data on the 3D dataset is remarkably significant. We obtained improvements of +16.8% ACC, +8.3% ACC, +9.8% ACC when using 1%, 10%, and 100% of the training data, respectively.

**Framework module** In this section, we further analyze the effects of self feature distillation and attentive slices selection on our framework. We conduct a linear classification task on downstream 2D datasets CheXpert and RSNA, as well as the 3D dataset CC-CCII. The results are summarized in Table 5. The experimental results show that incorporating both self feature distillation and attentive slices selection into our framework significantly improves the performance across all data splits and datasets.

## 5 CONCLUSION

In this paper, we propose a novel approach called *UniMedI* that leverages diagnostic reports as a shared semantic space to create unified representations for diverse modalities of medical images, with a specific emphasis on 2D and 3D images. By using medical diagnostic reports as a bridge, we establish the unified vision-language framework that connects visual medical data across different modalities. Moreover, with the guidance of the text, we effectively extract visual modality information and accurately identify affected areas in 2D images and lesion slices in 3D CT scans, thereby enhancing consistency across various visual data modalities. Extensive experiments demonstrate *UniMedI*'s superior performance in these downstream tasks(classification, segmentation, and retrieval) on various 2D and 3D medical image datasets. We hope our work can promote the exploration of VLP in medical image processing.

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

# A  MORE IMPLEMENTATION DETAILS OF PRE-TRAINING

## A.1  IMPLEMENTATION DETAILS

Following Gloria Huang et al. (2021), we utilize ViT-B/16 Dosovitskiy et al. (2020) as the vision encoder to extract representations in the common feature space for 2D and 3D visual data. We use BioClinicalBERT Alsentzer et al. (2019) as the text encoder to obtain the report embeddings. The vision encoder and text encoder are universal among 2D X-rays and 3D CT scans data. It is worth noting that the patch embed module of vision encoder has different operations for 2D X-rays and 3D CT scans. In general, the image size of 2D images is $224 \times 224$ and the volume size of 3D volumes is $128 \times 128 \times 32$. We pre-train our *UniMedI* framework 50 epochs on 8 pieces of Tesla V100 GPUs with batch size of 128. The optimizer is AdamW Loshchilov & Hutter (2017) with learning rate of $2e^{-5}$ and weight decay of 0.05, where the learning rate follows a linear warmup with cosine annealing scheduler Loshchilov & Hutter (2016). We initialize learning rate as $1e^{-8}$ and warmup epoch as 20.

# B  MORE IMPLEMENTATION DETAILS OF DOWNSTREAM TASKS

## B.1  MEDICAL CLASSIFICATION

**2D Medical Image Classification.** Except for the fine-tuning of the entire CheXpert dataset, where we use a batch size of 96, we use a batch size of 48 for the rest of the linear classification settings. Similar to the image preprocessing of MIMIC-CXR, we resize the larger dimension to 256 and pad zeros on the smaller side, resulting in an image size of $256 \times 256$. Then, we randomly crop (for training) or centered crop (for validation and testing) an image to $224 \times 224$ and normalize it into the range [0, 1] as the input for the model. The optimizer used is AdamW Loshchilov & Hutter (2017) with a learning rate of 5e-4 (except for COVIDx where we use 5e-3) and weight decay of 1e-6. We fine-tune the image classifier for 50 epochs and implement early stopping when the validation loss does not decrease for 10 consecutive runs. Afterward, we save the checkpoint model with the lowest validation loss for testing.

**3D Medical Image Classification.** (1) CC-CCII Zhang et al. (2020) contains a total number 617,775 slices of 6,752 CT scans from 4,154 patients. The task is to classify each volume into three categories: *novel coronavirus pneumonia*, *common pneumonia*, and *normal*. We use a batch size of 8. We resize the 3D volumes to $32 \times 128 \times 128$. We use the randomflip to augment the train set. The optimier used is AdamW and we train the classifier for 50 epochs. (2) LUNA 16 Setio et al. (2017), which is established from LIDC-IDRI Armato III et al. (2011). It finally contains 888 CT scans with annotations which removes CT scans with slice thickness greater than 3mm of LIDC-IDRI database. The task is a binary classification, *i.e.*, classifying each CT volume into *pulmonary nodule* or *normal*. The optimier used is AdamW and we train the whole network for 100 epochs. Our baseline methods include UniMiss Xie et al. (2022) and Joint Nguyen et al. (2023), which belongs to 2D and 3D co-learning methods. Unimiss not only learns 2D and 3D representations but also concurrently learn all 2D sections derived from 3D volumes, along with all 2D X-ray data. Joint directly learn all 2D sections derived from 3D volumes, along with all 2D X-ray data through contrastive learning.

## B.2  MEDICAL SEGMENTATION.

**2D Medical Image Segmentation.** In the case of the RSNA dataset, we create masks for the pneumonia-affected areas based on the provided bounding boxes. These images and corresponding masks are then resized to dimensions of $224 \times 224$. To augment the training set, we implement ShiftScaleRotate, encompassing random affine transformations such as translation, scaling, and rotation. Following this, the images are normalized to fall within the [0, 1] range before being supplied to the semantic segmentation model. we use the SETR-PUP (progressive upsample) architecture in Zheng et al. (2021) by replacing the encoder with *UniMedI*. We freeze the pre-trained image encoder and only train decoder portion. The training process involves the use of the AdamW optimizer with a learning rate of 5e-4 and a weight decay of 1e-6. As suggested by Huang et al. (2021), we adopt a combined loss equation of $\alpha \times$ FocalLoss + DiceLoss, with $\alpha$ set to 10. The semantic segmentation model undergoes fine-tuning for 50 epochs, with batch size 16 and early stopping im-

| Method | RSNA | | |
| --- | --- | --- | --- |
| | 1% | 10% | 100% |
| ConVIRT | 8.2 | 5.6 | 17.9 |
| GLoRIA | 9.8 | 14.8 | 18.8 |
| GLoRIA-MIMIC | 11.6 | 16.1 | 24.8 |
| MGCA (ResNet-50) | 12.9 | 16.8 | 24.9 |
| MGCA (ViT-B) | 14.7 | 18.4 | 25.8 |
| *UniMedI* | **15.5** | **19.2** | **26.6** |

Table 8: Object detection results (mAP [%]) on RSNA. Each dataset is fine-tuned with 1%, 10%, 100% training data. Best results are in boldface.

| Method | AMOS | | |
| --- | --- | --- | --- |
| | 20% | 40% | 100% |
| UniMiss | **79.5** | 82.3 | 85.8 |
| *UniMedI* | 78.8 | **82.9** | **86.4** |

Table 9: 3D Semantic Segmentation results on AMOS (Dice [%]). AMOS is fine-tuned with 20%, 40%, 100% training data. Best results are in boldface.

plemented if the validation loss ceases to decrease after 10 consecutive runs. The checkpoint model that exhibits the minimum validation loss is then preserved for testing.

**3D Medical Image Segmentation.** In the case of the BCV dataset, the images and correspoinding. The 3D volumes are resized to $48 \times 224 \times 224$. To augment the traning set, we implement random rotation, scaling, flipping, adding white Gaussian noise, Gaussian blurring, adjusting rightness and contrast, simulation of low resolution, and Gamma transformation. We use the UN-ETR Hatamizadeh et al. (2022) architecture by replace the encoder with pre-trained *UniMedI*. We freeze the pre-trained image encoder and only train decoder portion. The training process involves the use of the AdamW optimizer with a learning rate of 1e-4. We adopt a combined loss equation of Dice + CE. The semantic segmentation model fintunes for 25,000 iterations with batch size 2.

## C MORE ANALYSIS

### C.1 PNEUMONIA DETECTION IN RSNA

We evaluate the localized performance of pre-trained image encoder on RNSA Pneumonia. RSNA contains 29700 frontal view radiograph. The task is to predict bounding boxes indicating evidence of pneumonia. Due to use ViT-B as our bakcbone, it is sufficient to build a simple feature pyramid from a single-scale feature map. Therefore, we evaluate the detection performance by ViTDet Li et al. (2022) with using the pre-trained ViT-B as a frozen backbone and only finetuning the non-backbone layers. Similarly, we finetune the model by 1%, 10% and 100% training data to evaluate the data efficiency.

### C.2 3D MEDICAL SEGMENTATION IN AMOS

AMOS is a large-scale, diverse, clinical dataset for abdominal organ segmentation, which is divided into 200/100 CTs for training/validation. We use the validation set as our test set and the training details is the same as B.2. We report the Dice score (%) training with 20%, 40%, and 100% portion.

### C.3 DIFFERENT METRICS IN COVIDx

We applied two distinct evaluation metrics, namely AUC (Area Under the Curve) and ACC (Accuracy), to assess the performance of our model on the COVIDx dataset. AUC is a widely used metric in machine learning and it represents the probability that a random positive example will be ranked higher than a random negative example. A higher AUC indicates better model performance. On the other hand, Accuracy (ACC) is a measure of how many predictions a model gets right out of all

| Method | COVIDx (Acc/AUC) | | |
|---|---|---|---|
| | 1% | 10% | 100% |
| MGCA | 74.8/89.0 | 84.8/97.0 | 92.3/97.9 |
| *UniMedI* | **80.3/93.5** | **92.4/98.1** | **94.6/98.1** |

Table 10: 3D Semantic Segmentation results on AMOS (Dice [%]). AMOS is fine-tuned with 20%, 40%, 100% training data. Best results are in boldface.

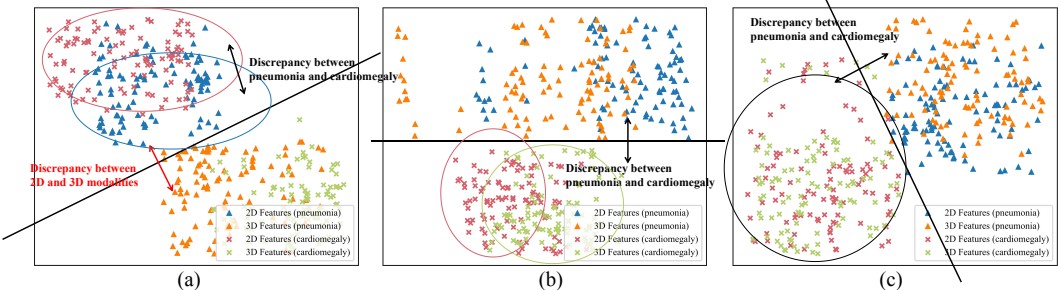

Figure 6: t-SNE visualizations of image representations by models trained with different methods. (a) Two models for different image modalities are trained individually in separate VLP process. (b) One models for different image modalities are trained in one VLP processes, but without designes in *UniMedI*. (c) *UniMedI*. We use circles to highlight differences between different images.

the predictions it makes. It is calculated as the number of correct predictions divided by the total number of predictions. The results of our evaluation using these metrics on the COVIDx dataset are presented in Table 10. These findings provide insights into the robustness of our model.

## C.4 VISUALIZATION OF FEATURE QUALITY

We add three t-SNE visualization in Figure. 6. Compared to Figure. 2, Figure. 6 add more class (cardiomegaly) to demonstrate the ability to unify different modal representations. We have marked the changes in distances between different modalities in the figure. As shown in Figure. 6, UniMedI effectively reduces the distance between different modalities and stablishes a unified representation space.

