# OpenReview forum: "Unified Medical Image Pre-training in Language-Guided Common Semantic Space"
_ICLR.cc/2024/Conference — Submitted to ICLR 2024_

### Official Review · Reviewer_vK4s · 2023-10-29

**Soundness:** 3 good
**Presentation:** 2 fair
**Contribution:** 2 fair
**Rating:** 6
**Confidence:** 4

**Summary:**

The paper introduces a visual-language pre-training method that can handle both 2D and 3D medical image data. It aligns different modalities of image data with their corresponding diagnostic reports and enhances the correlation between different modalities using MIM-based self-distillation.

**Strengths:**

This paper is well-organized and clearly described, and the figures are intuitive. Creating a unified model that can effectively handle various kinds of image data is a valuable problem. The strengths of the paper include:

1. The paper is well-written. The organization is clear, and the paper is easy to follow.
2. The studied problem is meaningful. The paper offers a unified framework to integrate multi-modal medical language-guided images into semantic space, facilitating the analysis and interpretation of complicated imaging data.
3. The approach obtains superior performance in the downstream classification and semantic segmentation tasks.

**Weaknesses:**

The weaknesses of the paper include:
1. The novelty appears to be somewhat incremental, as the method may seem to be a direct application of Attentive Mask CLIP proposed by Yang et al. (2023).
2. Although the method is capable of handling 3D image data, it treats them as separate 2D images and does not consider the structural aspects of 3D data.
3. Some implementation details need to be further explained. For instance, the structures of the segmentation decoder.

**Questions:**

1. As mentioned above, Please provide the decoder structure used for the segmentation task.
2. The medical semantic segmentation task on the RSNA and BCV datasets appears to involve predicting bounding boxes that indicate evidence of pneumonia, thus making it more akin to a detection problem rather than a segmentation problem.
3. Please elucidate the distinctions between the proposed method and Attentive Mask CLIP proposed by Yang et al. (2023).

---

> ### Author Response · Authors · 2023-11-23
> **Response to Reviewer vK4s (Part 1)**
>
> We thank the reviewer for the insightful comments, and we address the questions below:
>
> >(**W1**). The novelty appears to be somewhat incremental, as the method may seem to be a direct application of Attentive Mask CLIP proposed by Yang et al. (2023).
>
> >(**Q3**) Please elucidate the distinctions between the proposed method and Attentive Mask CLIP proposed by Yang et al. (2023).
>
>
> We re-elaborate more here on the novelty of our method, which is quite different from Attentive Mask CLIP (ACLIP).
>
> First, we target an unified VL framework capable of handling various medical modalities (e.g. 2D X-rays, 3D CT), and Figure 2 shows that merging representations of different medical image modalities into a language-guided common semantic space is challenging. Naively using language guidance and VL contrastive learning (Figure 2a) and simply unifying in one model (Figure 2b) cannot achieve the goal of merging. To this end, we introduce several designs for UniMedI, and *all of the designs aim at better merging 2D and 3D medical images into a language-guided common semantic space*. To best of our knowledge, we are the first one merging 2D and 3D medical images into a unified VL framework (we also provide a table to summarize the difference between our work and existing methods in this field at the end of this answer). Particularly, ACLIP cannot handle multi-modal images, and it can only be used for single modal images. So, *ACLIP cannot be directly applied to our problem*.
>
>
> The first design is incorporating the guidance from report via VL contrastive learning. Generally, the $[CLS]$ token in the image side is optimized with representation of the paired report. And this image $[CLS]$ token is used in the following parts of UniMedI to ensure that the integrating is guided by language. We also have a novel motivating observation here for our problem (shown in Figure 1 and introduction). We see that despite big differences, medical images from various modalities share a common semantic latent space, which captures the underlying features of an individual's health status, and such status are reflected in medical reports via language. *This motivating observation is insightful in our problem and has not been mentioned in previous works as far as we know.*
>
> The second design is our pipeline. Since no paired 2D and 3D image data exists, so we first select informative 2D slice from 3D volume as the bridge via the proposed attentive selection strategy. And then 2D slices, 2D X-ray and 3D volume are sent to the vision encoder together, enabling better merging. *ACLIP does not have this pipeline and the attentive selection strategy, since bridging multimodal images is not handled in ACLIP.* We have updated our paper by including a subsfigure in Figure 3 to show this pipeline.
>
> The third design is the auxiliary task of mask and recovery implemented by self-distillation. Since 2D and 3D tokens are sent to the visual encoder at the same time and a large proportion of tokens are masked, this auxiliary task can help to enhance interactions among 2D and 3D tokens, facilitating merging representations. *The mask in ACLIP is used for removing redundant information, while the auxiliary task in UniMedI is enhancing dimensional interactions.*
>
> Besides, the vision encoder has *separate tokenizer for 2D and 3D data and a shared backbone*, which is also designed for representation integration, and not used by ACLIP.
>
> In summary, the problem we addressed in this paper is quite novel, and all designs in UniMedI aim at better solving the targeted problem. Maybe some losses are similar with other methods, but they are introduced with different purpose here, and thus brings different effects. And there are lots of difference compared with ACLIP, as we listed above.
>
> | Method | Vision-Language | Unify | Downstream Task | Input Medical Image type |
> | :------: | :---------------: | :-----: | :---------------: | :------------------------: |
> |COVIRT [1] | $\checkmark$ | --- | Classification, Segmentation, Detection | 2D Images |
> |GLoRIA-MIMIC [2] | $\checkmark$ | --- | Classification, Segmentation, Detection | 2D Images |
> |MGCA [3] | $\checkmark$ | --- | Classification, Segmentation, Detection | 2D Images |
> | Joint [4]   | ---| $\checkmark$| Classification, Segmentation  | 2D Images, 2D slices  |
> | UniMiss [5] | ---| $\checkmark$| Classification, Segmentation  | 2D Images, 3D volumes, 2D slices |
> | UniMedI     | $\checkmark$ | $\checkmark$| Classification, Segmentation, Detection(Appendix)          | **2D Images, 3D volumes, 2D slices** |

---

> > ### Author Response · Authors · 2023-11-23
> > **Response to Reviewer vK4s (Part 2)**
> >
> > >(**W2**). Although the method is capable of handling 3D image data, it treats them as separate 2D images and does not consider the structural aspects of 3D data.
> >
> > Actually, the structural aspects of 3D data is considered in UniMedI. Specifically, after extracting 2D slices, the 3D volume and 2D slices are sent to the vision encoder as inputs at the same time. With separate tokenizers, 2D and 3D tokens exists in one pass. In this way, we can use the mask and recovery task to enhance dimensional interactions. Sorry for causing confusion. We have revised Figure 3 by incorporating a subfigure to illustrate the overall pipeline, and adjust corresponding descriptions, making this point clear.
> >
> >
> >
> >
> >
> > >(**W3**) Some implementation details need to be further explained. For instance, the structures of the segmentation decoder.
> >
> > >(**Q1**) As mentioned above, Please provide the decoder structure used for the segmentation task.
> >
> > Thanks for the suggestion. We have added more implementation details in Appendix B, including the structures of the segmentation decoder. Generally, we use SETR-PUP [6] for 2D segmentation tasks and UNETR [7] for 3D segmentation tasks.
> >
> >
> > >(**Q2**) The medical semantic segmentation task on the RSNA and BCV datasets appears to involve predicting bounding boxes that indicate evidence of pneumonia, thus making it more akin to a detection problem rather than a segmentation problem.
> >
> >
> > Generally, predicting bounding boxes is involved for the RSNA dataset, but not for BCV dataset. RSNA is a pneumonia dataset, and detection of bounding boxes can serve as a preliminary step in the segmentation process. So, we add results for segmentation tasks in RSNA in the below table and Appendix C.1 (metric: Dice[%]). Note that comparing with detecting bounding boxes, segmentation can be a more complex task providing more information via precisely segmenting the affected regions within the lungs. Moreover, segmentation task allows for a more detailed comparison with previous methods. So we listed segmentation results in our main paper.
> >
> > |         | 1%   | 10%  | 100% |
> > | :-------: | :----: | :----: | :----: |
> > | ConVIRT [1] | 8.2 | 5.6 | 17.9 |
> > | GLoRIA [2] | 9.8 |  14.8  | 18.8 |
> > |GLoRIA-MIMIC [2] | 11.6 | 16.1 | 24.8 |
> > | MGCA (ResNet-50) [3] | 12.9 | 16.8 | 24.9 |
> > | MGCA (ViT-B) [3] | 14.7 | 18.4 | 25.8 |
> > | UniMedI (ViT-B) | **15.5**  | **19.2**  | **26.6**  |
> >
> > Besdies, BCV is a organ segmentation dataset, but not a pneumonia dataset. Organ segmentation is a more meaningful task since detailed shape information of organs is useful in clinical problems, rather than bounding boxes of organs. Also, detecting is not involved for organ segmentation in common practice. So, this BCV dataset is only used to evaluate performance of organ segmentation, and no previous work uses it for detection. To further demonstrate the capability of our pretrained model on 3D data, we add a larger 3D segmentation dataset AMOS. Results is listed below and included in Appendix C.2.
> >
> > |         | 20%   | 40%  | 100% |
> > | ------- | ---- | ---- | ---- |
> > | UniMiss [5]    | 79.5 | 82.3 | 85.8 |
> > | UniMedI | **78.8** | **82.9** | **86.4** |
> >
> >
> > [1] Y. Zhang, H. Jiang, Y. Miura, C. D. Manning, and C. P. Langlotz. Contrastive learning of medical visual representations from paired images and text. arXiv preprint arXiv:2010.00747, 2020.
> >
> > [2] S.-C. Huang, L. Shen, M. P. Lungren, and S. Yeung. Gloria: A multimodal global-local
> > representation learning framework for label-efficient medical image recognition. In Proceedings
> > of the IEEE/CVF International Conference on Computer Vision, pages 3942–3951, 2021.
> >
> > [3] Fuying Wang, Yuyin Zhou, Shujun Wang, Varut Vardhanabhuti, and Lequan Yu. Multi-granularity
> > cross-modal alignment for generalized medical visual representation learning. Advances in Neural Information Processing Systems, 35:33536–33549, 2022.
> >
> > [4] Nguyen D M H, Nguyen H, Mai T T N, et al. Joint self-supervised image-volume representation learning with intra-inter contrastive clustering[C]//Proceedings of the AAAI Conference on Artificial Intelligence. 2023, 37(12): 14426-14435.
> >
> > [5] Xie Y, Zhang J, Xia Y, et al. Unimiss: Universal medical self-supervised learning via breaking dimensionality barrier[C]//European Conference on Computer Vision. Cham: Springer Nature Switzerland, 2022: 558-575.
> >
> > [6] Sixiao Zheng, Jiachen Lu, Hengshuang Zhao, Xiatian Zhu, Zekun Luo, Yabiao Wang, Yanwei
> > Fu, Jianfeng Feng, Tao Xiang, Philip HS Torr, et al. Rethinking semantic segmentation from
> > a sequence-to-sequence perspective with transformers. In Proceedings of the IEEE/CVF conference on computer vision and pattern recognition, pp. 6881–6890, 2021.
> >
> > [7] Ali Hatamizadeh, Yucheng Tang, Vishwesh Nath, Dong Yang, Andriy Myronenko, Bennett Landman, Holger R Roth, and Daguang Xu. Unetr: Transformers for 3d medical image segmentation.
> > In Proceedings of the IEEE/CVF winter conference on applications of computer vision, pp. 574–
> > 584, 2022.

---

### Official Review · Reviewer_8U1f · 2023-10-31

**Soundness:** 3 good
**Presentation:** 3 good
**Contribution:** 3 good
**Rating:** 6
**Confidence:** 4

**Summary:**

The UniMedI framework presents an innovative strategy for unifying the processing of diverse medical image modalities, especially 2D and 3D images, by employing diagnostic reports as a common semantic foundation. This approach facilitates the creation of consistent representations for different types of medical images. By harnessing the guidance provided by text, UniMedI excels in extracting pertinent visual information. It adeptly identifies impacted areas in 2D X-ray images and locates slices with lesions in more intricate 3D CT scans, thereby markedly improving coherence across various medical imaging formats.

**Strengths:**

The topic is new and clinically relevant.
It successfully develops unified representations for a variety of medical images, notably addressing both 2D and 3D image formats.
The framework's performance is thoroughly evaluated across 10 different datasets, encompassing a broad spectrum of medical imaging tasks including classification, segmentation, and retrieval.
The paper is well-structured and clearly written, making it accessible and easy to understand for readers.

**Weaknesses:**

The framework appears to be a multi-stage learning process rather than a true end-to-end 2D/3D multi-modal learning framework. It seems to involve selecting high-attention slices from 3D images and then training a 2D image-text encoder. If this understanding is correct, it is not as useful as a unified 2D/3D multi-modal learning framework.

While the use of t-SNE for problem definition is interesting, the paper lacks a comparative t-SNE plot in the results section to illustrate the impact of 2D and 3D co-learning. This could have provided clearer visual evidence of the model's effectiveness.

The paper does not explore straightforward alternative approaches, such as using a DENO or MAE, for learning all 2D sections from 3D volumes in conjunction with all 2D X-ray data together in a single self-supervised learning model.

There is a lack of clarity on why certain metrics, like AUC (Area Under Curve) and ACC (Accuracy), are chosen and reported in different sections. Particularly, I would suggest to use AUC over ACC in COVIDx, which could affect the clarity of the performance evaluation.

Many of the results, as shown in tables such as Table 4 (AUC), Table 5, and Table 7, indicate only marginal improvements. This raises questions about the practical significance and real-world applicability of the proposed framework.

**Questions:**

See the Weakness section. My further final decision will be decided based on the author's rebuttal.

---

> ### Author Response · Authors · 2023-11-23
> **Response to Reviewer 8U1f  (Part 1)**
>
> We thank the reviewer for the detailed feedback on our paper!
>
> >(**W1**) The framework appears to be a multi-stage learning process rather than a true end-to-end 2D/3D multi-modal learning framework. It seems to involve selecting high-attention slices from 3D images and then training a 2D image-text encoder. If this understanding is correct, it is not as useful as a unified 2D/3D multi-modal learning framework.
>
> We apologize for causing misunderstanding here. UniMedI is not a multi-stage learning process and it takes 2D slices and 3D images at the same time. Specifically, after extracting 2D slices, the 3D volume and 2D slices are sent to the vision encoder as inputs at the same time. With separate tokenizers, 2D and 3D tokens exists in one pass. In this way, we can use the mask and recovery task to enhance dimensional interactions. We do not have separate training stage for attentive slice selection and the whole model is updated with backprogration once each batch. To clarify more clearly, we provide explanations of UniMedI's pipeline below. In our revised paper, we have included a subfigure in Figure 3 to illustrate, and also adjusted corresponding descriptions.
>
> ### Pipeline of UniMedI
> As we have described in the first paragraph of Section 3, to overcome the challenges that no paired 2D and 3D image data exists, during the training process, UniMedI first extracts a portion of informative 2D slices from a 3D volume according to language guidance via the proposed **attentive slice selection strategy (shown in Figure 4)**. This creates 2D-3D data pairs to bridge the disparity between the two modalities. Then, selected 2D slices and 2D X-rays, together with original 3D data, are sent to **a unified vision encoder (shown in Figure 3)** to obtain image representations.
>
> Overall, UniMedI is a vision-language pre-training framework. The paired report of each X-ray/CT is encoded by the text encoder, and VL contrastive learning (shown in Figure 3) are employed to obtain a language-guided common semantic space for both 2D and 3D medical images.
>
> >(**W2**) While the use of t-SNE for problem definition is interesting, the paper lacks a comparative t-SNE plot in the results section to illustrate the impact of 2D and 3D co-learning. This could have provided clearer visual evidence of the model's effectiveness.
>
> We would like to clarify that the impact of 2D and 3D co-learning is already shown in Figure 2. Generally, UniMedI target the goal of mapping data from various medical image modalities into the shared semantic space, which is guided by language in reports, and Figure 2 (visualizations of representations) is used to show that this goal is hard to achieve. We provide the information in Figure 2 in the following table. In summary, Figure 2 shows that naively using language guidance (2a) and simply unifying in one model (2b) cannot achieve the our challenging goal. In contrast, UniMedI, which involves 2D and 3D co-learning, can better integrate different medical image modalities.
>
> |  | How the models are trained | What information is shown |
> | -------- | -------- | -------- |
> | Figure 2a    |   Two models for 2D and 3D images are trained individually in two VLP processes, respectively.   |  There is a **significant gap** in the representation of 2D and 3D medical modality data.    |
> | Figure 2b    |  One model for 2D and 3D images are trained in one VLP process.       |    The gap between the modalities have been **narrowed compared with 2a**, but overall, the 2D X-ray representations are distributed in the upper part of the figure, while the 3D CT representations are distributed in the lower part of the figure.      |
> | Figure 2c    |  One model for 2D and 3D images are trained in UniMedI, where the connection between image modalities are largely enhanced.        |    **The representations of different modalities are distributed together and closer compared with 2b**.
>
> Moreover, per suggestion from Reviewer zUjR, **we have added visualization of data representations of two different classes (i.e., pneumonia and cardiomegaly) in Figure 6 in Appendix C.4**. As can be seen, *even data coming from different classes, UniMedI can bring together representations for medical data of different modalities, while not making features of all classes similar, well preserving semantic information from reports in the representation space*. This nicely demonstrates the effectiveness of UniMedI on integrating medical multi-modal images into a language-guided common semantic space.

---

> > ### Author Response · Authors · 2023-11-23
> > **Response to Reviewer 8U1f (Part 2)**
> >
> > >(**W3**) The paper does not explore straightforward alternative approaches, such as using a DENO or MAE, for learning all 2D sections from 3D volumes in conjunction with all 2D X-ray data together in a single self-supervised learning model.
> >
> > First, we would like to clarify that the method reviewer mentioned (i.e., learning all 2D sections from 3D volumes in conjunction with all 2D X-ray data together in a single self-supervised learning model) is done by baselines we have compared. UniMiss [1] supports 2D images, 2D slices, and 3D volumes and the Joint method [2] directly converts 3D data into 2D slices. Both of them use contrastive learning for self-supervised learning.
> >
> > To further address your concerns, we have added other self-supervised methods in the medical field for comparison, including SimMIM [3] and DiRA [4]. SimMIM is mainly designed for 2D data, while DiRA proposes a pre-training strategy for both 2D and 3D data, but with different backbones. Results are in the below table (CheXpert$^\ast$ denoting full backbone fine-tuning and CheXpert$^\diamond$ representing linear classification, only fine-tuning the classification head). It is evident that UniMedI outperforms all other methods, even without fine-tuning for the evaluation on the large-scale CheXpert dataset.
> >
> > | Method                                                     | Downstream Task                                   | Performance                            | Input Medical Image type         |
> > | ---------------------------------------------------------- | ------------------------------------------------- | -------------------------------------- | -------------------------------- |
> > | SimMIM [3]                                                 | Classification                                    | CheXpert$^\ast$ (88.07)                | 2D Images                        |
> > | DiRA[4] **designed separate backbones for 2D and 3D data** | Classification, Segmentation | CheXpert$^\ast$ (87.59) | 2D Images, 3D volumes|
> > | UniMedI                                                    | Classification, Segmentation, Detection           | **CheXpert$^\diamond$ (90.50)**  | 2D Images, 3D volumes, 2D slices |
> >
> >
> > Moreover, we would like to recall the motivation of using vision-language contrastive learning and the example in Figure 1. Figure 1 indicates that despite big differences, medical images from various modalities share a common semantic latent space, which captures the underlying features of an individual's health status, and such status are reflected in medical reports via language. This observation motivate us to map data from various medical image modalities into the shared semantic space, which is guided by language in reports. Experiment results also verify that vision-language pre-training bring benefits for many downstream tasks.
> >
> > >(**W4**) There is a lack of clarity on why certain metrics, like AUC (Area Under Curve) and ACC (Accuracy), are chosen and reported in different sections. Particularly, I would suggest to use AUC over ACC in COVIDx, which could affect the clarity of the performance evaluation.
> >
> > We select these metrics generally by following previous works and thus can better compare with them. Per your suggestion, we provide both ACC and AUC (ACC/AUC) in the below table, and revise our manuscript accordingly to report the AUC for the COVIDx dataset in Appendix C.3.
> >
> > |         | 1%   | 10%  | 100% |
> > | ------- | ---- | ---- | ---- |
> > | MGCA [5]    | 74.8/89.0  | 84.8/97.0 | 92.3/97.9 |
> > | UniMedI | **80.3/93.5** | **92.4/98.1** | **94.6/98.1** |

---

> > > ### Author Response · Authors · 2023-11-23
> > > **Response to Reviewer 8U1f (Part 3)**
> > >
> > > >(**W5**) Many of the results, as shown in tables such as Table 4 (AUC), Table 5, and Table 7, indicate only marginal improvements. This raises questions about the practical significance and real-world applicability of the proposed framework.
> > >
> > > >>5.a. Table 4 (AUC), Table 5.
> > >
> > > Thank you for your insightful comments. We would like to first clarify that Tables 4 and 5 present our ablation studies, which is not used for demonstrating the practical significance of UniMedI. The effectiveness of UniMedI is shown via comparison with other methods in Table 1~3.
> > >
> > > Second, we want to emphasize that the CheXpert and RSNA datasets are quite challenging, and improving performance on these datasets is difficult. In previous work, Glora-MIMIC [6] only improved by 0.7 (CheXpert 100%) compared to ConVIRT [7], and MGCA [5] (ResNet50) only improved by 0.2 (CheXpert 100%) compared to Glora-MIMIC [6], with performance decreasing on RSNA (10, 100%). Considering these results, the improvement in Table 5 is quite significant on the CheXpert and RSNA datasets, where previous methods found it difficult to achieve the 1% improvement. Also, based on these results and Table 1~3, we can conclude that UniMedI achieved a slight improvement on complex 2D datasets and a significant improvement on 3D datasets.
> > >
> > > >>5.b. Table 7
> > >
> > > Thank you for your comments. You're correct that the performance increase shown in Table 5 is modest. The reason for this is that *the model structure of UniMiss [1] is a uniquely designed UNet, while ours is a more generic Vision Transformer (ViT)*. When transferring to downstream tasks, we can only achieve a slight improvement over UniMiss because the decoder of the UniMiss model structure has already been pre-trained, while ours is not.
> > >
> > > Furthermore, the BCV dataset used in this comparison is relatively small. To address your concern and provide more solid results, *we have included an additional larger dataset, AMOS*. This should give a better indication of how our model compares with UniMiss when trained with larger datasets. The results are shown in the following table:
> > >
> > >
> > > |         | 20%   | 40%  | 100% |
> > > | ------- | ---- | ---- | ---- |
> > > | UniMiss [1]    | 79.5 | 82.3 | 85.8 |
> > > | UniMedI | **78.8** | **82.9** | **86.4** |
> > >
> > > We appreciate your feedback and hope this explanation will help clarify our model's performance.
> > >
> > >
> > > [1] Xie Y, Zhang J, Xia Y, et al. Unimiss: Universal medical self-supervised learning via breaking dimensionality barrier[C]//European Conference on Computer Vision. Cham: Springer Nature Switzerland, 2022: 558-575.
> > >
> > > [2] Nguyen D M H, Nguyen H, Mai T T N, et al. Joint self-supervised image-volume representation learning with intra-inter contrastive clustering[C]//Proceedings of the AAAI Conference on Artificial Intelligence. 2023, 37(12): 14426-14435.
> > >
> > > [3] Ma, DongAo, Mohammad Reza Hosseinzadeh Taher, Jiaxuan Pang, Nahid UI Islam, Fatemeh Haghighi, Michael B. Gotway, and Jianming Liang. "Benchmarking and boosting transformers for medical image classification." In MICCAI Workshop on Domain Adaptation and Representation Transfer, pp. 12-22. Cham: Springer Nature Switzerland, 2022.
> > >
> > > [4] Haghighi, Fatemeh, Mohammad Reza Hosseinzadeh Taher, Michael B. Gotway, and Jianming Liang. "DiRA: Discriminative, restorative, and adversarial learning for self-supervised medical image analysis." In Proceedings of the IEEE/CVF Conference on Computer Vision and Pattern Recognition, pp. 20824-20834. 2022.
> > >
> > > [5] Fuying Wang, Yuyin Zhou, Shujun Wang, Varut Vardhanabhuti, and Lequan Yu. Multi-granularity
> > > cross-modal alignment for generalized medical visual representation learning. Advances in Neural
> > > Information Processing Systems, 35:33536–33549, 2022.
> > >
> > > [6] S.-C. Huang, L. Shen, M. P. Lungren, and S. Yeung. Gloria: A multimodal global-local
> > > representation learning framework for label-efficient medical image recognition. In Proceedings
> > > of the IEEE/CVF International Conference on Computer Vision, pages 3942–3951, 2021.
> > >
> > > [7] Y. Zhang, H. Jiang, Y. Miura, C. D. Manning, and C. P. Langlotz. Contrastive learning of medical visual representations from paired images and text. arXiv preprint arXiv:2010.00747, 2020.

---

### Official Review · Reviewer_EXCm · 2023-10-31

**Soundness:** 3 good
**Presentation:** 3 good
**Contribution:** 3 good
**Rating:** 6
**Confidence:** 5

**Summary:**

This paper proposed a pre-training method to jointly learn 2D and 3D medical images with radiology reports. The method uses diagnostic reports as a common semantic space to create unified representations for 2D X-ray and 3D CT scans. To jointly incorporate 2D and 3D data, an attentive slice selection method was designed to select the disease-relevant 2D slices from 3D CT scans.

**Strengths:**

- The proposed UniMedI framework is designed to handle different imaging modalities (e.g., 2D X-rays, 3D CT scans), which is significant because medical imaging is inherently diverse, and most existing models are limited to single-dimension data.
- By using associated diagnostic reports, the framework can tap into the rich semantic information that is typically underutilized in image-only models, potentially leading to more context-aware representations.

**Weaknesses:**

- The description of the framework is not clear. It would be better if you could mark the (1)-(4) in Figure 3.

- Based on the methodology description, it seems that the proposed framework is a combination of the existing methods except for the attentive slice selection. The motivation for introducing the auxiliary task is not clear. Directly saying “inspired by…” is not a good writing style. Please explicitly present the main motivation of all your methodology components.

- The medical classification experiments are weak since most selected datasets are old and many other dedicated methods already achieved great performance. Please use the latest RSNA datasets for validation as well.

- For the 3D segmentation tasks, the small BCV dataset cannot provided statistically significant results (other previous works also used this BCV dataset is not a good excuse). Please use larger datasets like MICCAI FLARE and AMOS.

**Questions:**

- The authors designed an attentive slice selection method to select the disease relevant 2D slices from 3D volume. What’s the accuracy of the method? It is not validate yet but it could be easily done by testing it on some lung/abdomen tumor CT datasets that have tumor segmentation masks, e.g., check whether the selected slices cover all the tumors and compute the accuracy.

- Sec 3. “selected 2D input is fed into the network along with the 3D tokens..”, You already covert the 3D images into 2D key slices. What are the 3D tokens here?

- Please explicitly clarify the unique motivations of EMA and auxiliary tasks rather than just saying "inspired by"

Minors: Please polish the writing:
Page 5. Sec. 3.3 “In Section 3.1…. Then in Section 3.1…”

---

> ### Author Response · Authors · 2023-11-22
> **Response to Reviewer EXCm**
>
> We thank the reviewer for providing insightful comments on our paper. We provide clarifications to the concerns below:
>
> >(**W1**) The description of the framework is not clear. It would be better if you could mark the (1)-(4) in Figure 3.
>
>
> Thank you for your valuable suggestion. We apologize if the description of our framework was unclear in the initial submission.  We have revised Figure 3 in our paper to give a clearer presentation without changing the content. The left of Figure 3 demonstrates the overall pipeline of UniMedI, and the right part of Figure 3 and Figure 4 present details on the modules used in the pipeline. Also, we have marked (1)-(4) in Figure 3 upon your request. To further address your concerns, we would like to briefly review UniMedI here.
>
> ### Pipeline of UniMedI
> As we have described in the first paragraph of Section 3, to overcome the challenges that no paired 2D and 3D image data exists, during the training process, UniMedI first extracts a portion of informative 2D slices from a 3D volume according to language guidance via the proposed **attentive slice selection strategy (shown in Figure 4)**. This creates 2D-3D data pairs to bridge the disparity between the two modalities. Then, selected 2D slices and 2D X-rays, together with original 3D data, are sent to **a unified vision encoder (shown in Figure 3)** to obtain image representations.
>
> Overall, UniMedI is a vision-language pre-training framework. The paired report of each X-ray/CT is encoded by the text encoder, and VL contrastive learning (shown in Figure 3) are employed to obtain a language-guided common semantic space for both 2D and 3D medical images.
>
> ### Key module 1: the unified vision encoder (Figure 3)
>
> For the vision encoder, first, 2D data (selected 2D slices and 2D X-rays) and 3D data (original 3D volumes) are processed by 2D tokenizer $T_{2D}$ and 3D tokenizer $T_{3D}$, respectively. Seperate tokenizers encode input data from different modalities into a series of tokens. 3D CT tokens and corresponding 2D slices tokens are concatenated together for further procesing by a shared backbone. *Note here input contains both 2D and 3D data, so we have 2D and 3D tokens at the same time.* 2D X-rays tokens directly processing by a shared backbone.
>
> Then, to enhance the cross-dimensional communication among 2D and 3D tokens, we adopt **an auxiliary task, i.e., mask and recovery**, and this task is implemented by the **self-distillation method**. Specifically, our visual encoder contains the teacher network $\overline{E}_v$ and the student network $E_v$, where $\overline{E}_v$ is updated by exponential moving averaged (EMA) over $E_v$. Tokens, once encoded by the tokenizer, are input into the teacher network. This process yields global and local embeddings and simultaneously outputs attention scores for each patch of the X-rays or CT scans, as outlined in **Equation 1** in main text. Depending on the scores of each patch, we mask the less significant areas (those with lower scores) for the input into the student network. Since **the student network only inputs important areas of the medical images**, the resulting visual encoding is more closely matched with the semantics included in the report. Finally, loss of the self-distillation method is applied to both the global $[CLS]$ token (i.e., $L_{icl}$) and local patch tokens (i.e., $L_{pcl}$).
>
> ### Key module 2: the attentive slices selection strategy (Figure 4)
>
> The aim of attentive slice selection is selecting informative 2D slices from the 3D CT volume. The 3D CT volume is processed by the tokenizer $T_{3D}$ and the teacher network, and thus, attention scores for patches in the CT can be obtained. Then, the scores of all tokens within one slice are averaged to derive the score of the corresponding slice. This process is referred to as the *Inter-Slice Average*, as outlined in **Equation 2** in main text. Slices with top $k$ score are selected as the connection between 2D and 3D data.
>
> Note that these scores are attention weights between the $[CLS]$ token and other patch tokens, and the $[CLS]$ token is used for VL contrastive learning. Therefore, this slice selection process is guided by text information in the report, and 2D slices containing important information in the report can be selected accordingly.

---

> > ### Author Response · Authors · 2023-11-22
> > **Response to Reviewer EXCm**
> >
> > >(**W2**) Based on the methodology description, it seems that the proposed framework is a combination of the existing methods except for the attentive slice selection. The motivation for introducing the auxiliary task is not clear. Directly saying “inspired by…” is not a good writing style. Please explicitly present the main motivation of all your methodology components.
> >
> >
> > To better explain the motivation of designs in our method, we first elaborate on what we have done in this paper. As we have stated in the introduction, we target expanding the data scale for medical vision language (VL) pre-training. Here, one obvious and important problem is that medical images have different dimensions. Therefore, we propose *UniMedI*, an unified VL framework capable of handling various medical modalities (e.g. 2D X-rays, 3D CT). However, Figure 2 shows that merging representations of different medical image modalities into a language-guided common semantic space is challenging. Naively using language guidance and VL contrastive learning (Figure 2a) and simply unifying in one model (Figure 2b) cannot achieve the goal of merging. To this end, we introduce several designs for UniMedI, and **all of the designs aim at better merging 2D and 3D medical images into a language-guided common semantic space**. To best of our knowledge, we are the first one merging 2D and 3D medical images into a unified VL framework (we also provide a table to summarize the difference between our work and existing methods in this field at the end of this answer).
> >
> > The first design is incorporating the guidance from report via VL contrastive learning. Generally, the $[CLS]$ token in the image side is optimized with representation of the paired report. And this image $[CLS]$ token is used in the following parts of UniMedI to ensure that the integrating is guided by language.
> >
> > The second design is our pipeline. As we have explained in the last question, since no paired 2D and 3D image data exists, so we first select informative 2D slice from 3D volume as the bridge via the attentive selection strategy. And then 2D slices, 2D X-ray and 3D volume are sent to the vision encoder together, enabling better merging.
> >
> > The third design is the auxiliary task of mask and recovery implemented by self-distillation. Since 2D and 3D tokens are sent to the visual encoder at the same time and a large proportion of tokens are masked, this auxiliary task can help to enhance interactions among 2D and 3D tokens, facilitating merging representations.
> >
> > Besides, the vision encoder has separate tokenizer for 2D and 3D data and a shared backbone, which is also designed for representation integration.
> >
> > In summary, the problem we addressed in this paper is quite novel, and all designs in UniMedI aim at better solving the targeted problem. Maybe some losses is similar with other methods, but they are introduced with different purpose here, and thus brings different effects. Moreover, we have changed the writing upon your suggestion in our paper.
> >
> >
> >
> >
> >
> > | Method | Vision-Language | Unify | Downstream Task | Input Medical Image type |
> > | :------: | :---------------: | :-----: | :---------------: | :------------------------: |
> > |COVIRT [1] | $\checkmark$ | --- | Classification, Segmentation, Detection | 2D Images |
> > |GLoRIA-MIMIC [2] | $\checkmark$ | --- | Classification, Segmentation, Detection | 2D Images |
> > |MGCA [3] | $\checkmark$ | --- | Classification, Segmentation, Detection | 2D Images |
> > | Joint [4]   | ---| $\checkmark$| Classification, Segmentation  | 2D Images, 2D slices  |
> > | UniMiss [5] | ---| $\checkmark$| Classification, Segmentation  | 2D Images, 3D volumes, 2D slices |
> > | UniMedI     | $\checkmark$ | $\checkmark$| Classification, Segmentation, Detection(Appendix)          | **2D Images, 3D volumes, 2D slices** |

---

> > > ### Author Response · Authors · 2023-11-22
> > > **Response to Reviewer EXCm**
> > >
> > > >(**W3**) The medical classification experiments are weak since most selected datasets are old and many other dedicated methods already achieved great performance. Please use the latest RSNA datasets for validation as well.
> > >
> > > Thank you for your feedback. Actually, we **have already use the latest RSNA datasets** (specifically, stage 2 data but not stage 1) for evaluation. The stage 2 dataset is larger than stage 1, and is adopted by latest previous works [2,3]. We apologize for the confusion and have revised the manuscript by explicitly saying that we use the stage 2 data of RSNA in Section 4.2.
> > >
> > > Besides the classification experiments, we also add the **segmentation** (Dice [%]) and **detection** tasks (mAP [%]) and compared them with the most advanced method MGCA, as shown in the following tables.
> > >
> > >
> > > Segmentation:
> > > |         | 1%   | 10%  | 100% |
> > > | ------- | ---- | ---- | ---- |
> > > | MGCA [3]    | 66.2 | 71.3 | 73.6 |
> > > | UniMedI | **67.8** | **73.1** | **75.3** |
> > >
> > > Detection:
> > > |         | 1%   | 10%  | 100% |
> > > | :-------: | :----: | :----: | :----: |
> > > | ConVIRT [1] | 8.2 | 5.6 | 17.9 |
> > > | GLoRIA [2] | 9.8 |  14.8  | 18.8 |
> > > |GLoRIA-MIMIC [2] | 11.6 | 16.1 | 24.8 |
> > > | MGCA (ResNet-50) [3] | 12.9 | 16.8 | 24.9 |
> > > | MGCA (ViT-B) [3] | 14.7 | 18.4 | 25.8 |
> > > | UniMedI (ViT-B) | **15.5**  | **19.2**  | **26.6**  |
> > >
> > > Above results are added to Appendix C.1 to further demonstrate the effectiveness of our approach.
> > >
> > >
> > >
> > >
> > > >(**W4**) For the 3D segmentation tasks, the small BCV dataset cannot provided statistically significant results (other previous works also used this BCV dataset is not a good excuse). Please use larger datasets like MICCAI FLARE and AMOS.
> > >
> > >
> > >
> > > In response to your suggestion, we add an experiment using a larger dataset AMOS. Results are listed below and this experiment is added to Apppendix C.2. We believe that this result can demonstrate the superiority of UniMedI with statistical significance.
> > >
> > > |         | 20%   | 40%  | 100% |
> > > | ------- | ---- | ---- | ---- |
> > > | UniMiss [5]    | 79.5 | 82.3 | 85.8 |
> > > | UniMedI | **78.8** | **82.9** | **86.4** |
> > >
> > >
> > >
> > > As for MICCAI FLARE, this dataset contains images of many other body parts besides the chest, but chest is the main focus of our pre-trained model due to that publicly available paired image-report datasets are mainly about chest. Adapting a model pre-trained with chest data to images of other body parts is non-trivial and beyond the scope of this paper. We may leave it as our future work.
> > >
> > >
> > > >(**Q1**) The authors designed an attentive slice selection method to select the disease relevant 2D slices from 3D volume. What’s the accuracy of the method? It is not validate yet but it could be easily done by testing it on some lung/abdomen tumor CT datasets that have tumor segmentation masks, e.g., check whether the selected slices cover all the tumors and compute the accuracy.
> > >
> > >
> > > Thanks for this constructive feedback, we have conducted an additional experiment following your suggestion. Specifically, we utilize the subset0 of LUNA16 dataset, a standard dataset in LUNA Nodule Detection, to verify whether the selected slices encompass the nodules. We filter the original CTs from 89 to 67 by selecting nodules larger than 3 mm. The input size of CT is 32 $\times$ 224 $\times$ 224. We see that if the nodules are included in the slices UniMedI selected. A continuous set of 32 slices that contain nodules are selected for this performance validation. We follow the LUNA challenge guidelines that a nodule is deemed to be within three slices near the central coordinate. We selected the top 20% of slices with the highest attentive scores.
> > >
> > > The final accuracy of UniMedI is 76.1%. For reference, accuracy of random choosing is 53.73%. Although the absolute accuracy value is not pretty high, the improvement compared with the random model is significant. Note that (1) no explicitly direct supervision regarding nodules and its position is used in UniMedI, and we only use the report and no extra processing for reports is involved; (2) no specific and direct learning strategy is used in UniMedI for extracting slices with nodules (attentive slice selection is an indirect one because it emphasize report information but not exact nodule slices). The selection is based on attention weights, which is automatically learned in an end-to-end way. In other words, this selection ability is a kind of "free lunch" brought by UniMedI, so it might not be fair if it is compared with some specifically designed methods.

---

> > > > ### Author Response · Authors · 2023-11-22
> > > > **Response to Reviewer EXCm**
> > > >
> > > > >(**Q2**) Sec 3. “selected 2D input is fed into the network along with the 3D tokens..”, You already covert the 3D images into 2D key slices. What are the 3D tokens here?
> > > >
> > > > The 3D tokens are generated by feeding 3D volume into the vision encoder and being processed by the 3D tokenizer. After 2D slices are selected from 3D volume, we have 2D and 3D data as input at the same time, so with separate tokenizers, 2D and 3D tokens exists in one pass. In this way, we can use the mask and recovery task to enhance dimensional interactions. Sorry for causing confusion. We have revised Figure 3 by incorporating a subfigure to illustrate the overall pipeline, and adjust corresponding descriptions, making this point clear.
> > > >
> > > >
> > > >
> > > > >(**Q3**) Please explicitly clarify the unique motivations of EMA and auxiliary tasks rather than just saying "inspired by".
> > > >
> > > >
> > > > EMA is a technique used in the self-distillation method, to accomplish the mask and recovery auxiliary task. And the motivation to this task is better merging presentations of 2D and 3D medical images, as we have explained in details in the response to **W2**. The way to use EMA has been explained in the response to **W1**, "Key module 1: the unified vision encoder (Figure 3)" part. We have revised all "inspired by" sentences to make our statement precise.
> > > >
> > > >
> > > >
> > > >
> > > >
> > > >
> > > > >(**Q4**) Minors: Please polish the writing: Page 5. Sec. 3.3 “In Section 3.1…. Then in Section 3.1…”.
> > > >
> > > > Fixed!
> > > >
> > > >
> > > > [1] Y. Zhang, H. Jiang, Y. Miura, C. D. Manning, and C. P. Langlotz. Contrastive learning of medical visual representations from paired images and text. arXiv preprint arXiv:2010.00747, 2020.
> > > >
> > > > [2] S.-C. Huang, L. Shen, M. P. Lungren, and S. Yeung. Gloria: A multimodal global-local
> > > > representation learning framework for label-efficient medical image recognition. In Proceedings
> > > > of the IEEE/CVF International Conference on Computer Vision, pages 3942–3951, 2021.
> > > >
> > > > [3] Fuying Wang, Yuyin Zhou, Shujun Wang, Varut Vardhanabhuti, and Lequan Yu. Multi-granularity
> > > > cross-modal alignment for generalized medical visual representation learning. Advances in Neural
> > > > Information Processing Systems, 35:33536–33549, 2022.
> > > >
> > > > [4] Nguyen D M H, Nguyen H, Mai T T N, et al. Joint self-supervised image-volume representation learning with intra-inter contrastive clustering[C]//Proceedings of the AAAI Conference on Artificial Intelligence. 2023, 37(12): 14426-14435.
> > > >
> > > > [5] Xie Y, Zhang J, Xia Y, et al. Unimiss: Universal medical self-supervised learning via breaking dimensionality barrier[C]//European Conference on Computer Vision. Cham: Springer Nature Switzerland, 2022: 558-575.

---

> > > > > ### Comment · Reviewer_EXCm · 2023-11-22
> > > > >
> > > > > Thanks for your detailed response. Most of my concerns have been addressed. I raised my score to 6.

---

### Official Review · Reviewer_zUjR · 2023-11-01

**Soundness:** 2 fair
**Presentation:** 2 fair
**Contribution:** 2 fair
**Rating:** 3
**Confidence:** 5

**Summary:**

This paper proposed a unified framework for pre-training, unifying 2D (X-ray) and 3D (CT) modalities. The unification is realized by introducing language embeddings, which align the features of two unpaired images, but with the same pathological condition (e.g., pneumonia), into a similar space. Results show improved transfer learning performance in both 2D and 3D medical imaging tasks when pre-trained with language (pathological reports).

**Strengths:**

+ Table 4 demonstrates that leveraging a composite of 2D and 3D datasets during pre-training enhances performance.  This ablation analysis is clear and important.
+ Introducing language-vision framework into pre-training appears promising direction since the availability of pathological reports along with X-ray and CT images.

**Weaknesses:**

- The motivation of the paper, bridging features in 2D and 3D, is not validated (see Q1).
- The details of the method are confusing (see Q2).
- Lack of baseline methods for 2D and 3D pre-training (see Q3).

**Questions:**

1. From Figure 1, I fail to observe (c) is particularly better than (b) in terms of feature quality. I understood that the authors try to present the features of pneumonia get closer between 2D and 3D images, but this needs features of another class (e.g., nodule) as a comparison. It is possible that the proposed UniMedI makes features of all classes similar. I think the discrepancy between 2D and 3D modalities could be much larger than that among classes.

2. The illustration of Figure 3 and Figure 4 is unclear to me. What are the light orange boxes used for? What is the operation after T_3D and T_2D? The main text in the method section is not corresponding to Figure 3 or Figure 4. It is unclear how Equations (1-2) were integrated into the framework.

3. As a paper for pre-training, the authors did not compare with the many existing pre-trained models, neither 3D or 2D. For 2D X-ray imaging, there are many pre-trained models publicly available [e.g., 1-3]. For 3D CT imaging, a variety of public models are also missing [e.g., 4-6].

4. The authors did not provide sufficient reference for the baseline methods presented in Table 1.

**Reference**

[1] Ma, DongAo, Mohammad Reza Hosseinzadeh Taher, Jiaxuan Pang, Nahid UI Islam, Fatemeh Haghighi, Michael B. Gotway, and Jianming Liang. "Benchmarking and boosting transformers for medical image classification." In MICCAI Workshop on Domain Adaptation and Representation Transfer, pp. 12-22. Cham: Springer Nature Switzerland, 2022.

[2] Yan, Ke, Jinzheng Cai, Dakai Jin, Shun Miao, Dazhou Guo, Adam P. Harrison, Youbao Tang, Jing Xiao, Jingjing Lu, and Le Lu. "SAM: Self-supervised learning of pixel-wise anatomical embeddings in radiological images." IEEE Transactions on Medical Imaging 41, no. 10 (2022): 2658-2669.

[3] Zhang, Xiaoman, Chaoyi Wu, Ya Zhang, Weidi Xie, and Yanfeng Wang. "Knowledge-enhanced visual-language pre-training on chest radiology images." Nature Communications 14, no. 1 (2023): 4542.

[4] Tang, Yucheng, Dong Yang, Wenqi Li, Holger R. Roth, Bennett Landman, Daguang Xu, Vishwesh Nath, and Ali Hatamizadeh. "Self-supervised pre-training of swin transformers for 3d medical image analysis." In Proceedings of the IEEE/CVF Conference on Computer Vision and Pattern Recognition, pp. 20730-20740. 2022.

[5] Haghighi, Fatemeh, Mohammad Reza Hosseinzadeh Taher, Michael B. Gotway, and Jianming Liang. "DiRA: Discriminative, restorative, and adversarial learning for self-supervised medical image analysis." In Proceedings of the IEEE/CVF Conference on Computer Vision and Pattern Recognition, pp. 20824-20834. 2022.

[6] Chen, Sihong, Kai Ma, and Yefeng Zheng. "Med3d: Transfer learning for 3d medical image analysis." arXiv preprint arXiv:1904.00625 (2019).

---

> ### Author Response · Authors · 2023-11-22
> **Response to Reviewer zUjR**
>
> Thanks for your constructive comments on our work. We appreciate the opportunity and have addressed all your concerns point by point as follow, and also revised our manuscript accordingly.
>
> > (**W1**) The motivation of the paper, bridging features in 2D and 3D, is not validated (see Q1).
>
> > (**Q1**) From Figure 2, I fail to observe \(c\) is particularly better than \(b\) in terms of feature quality. I understood that the authors try to present the features of pneumonia get closer between 2D and 3D images, but this needs features of another class (e.g., nodule) as a comparison. It is possible that the proposed UniMedI makes features of all classes similar. I think the discrepancy between 2D and 3D modalities could be much larger than that among classes.
>
> First, in the revised version, we have highlighted the differences among each subfigures of Figure 2 as much as possible. In 2(b), overall, the 2D X-ray representations are distributed in the upper part of the figure, while the 3D CT representations are distributed in the lower part of the figure. In 2\(c\), representations of 2D and 3D images are generally merged together and closer compared with 2(b). In summary, Figure 2 shows that naively using language guidance (2a) and simply unifying in one model (2b) cannot achieve the our challenging goal, which is using text information in reports as the guidance to map data from various medical image modalities into the shared semantic space. In contrast, UniMedI can better integrate different medical image modalities.
>
> Also, to further address your concerns, **we have visualized data representations of two different classes (i.e., pneumonia and cardiomegaly) in Figure 6 in Appendix C.4**. As can be seen, *even data coming from different classes, UniMedI can bring together representations for medical data of different modalities, while not making features of all classes similar, well preserving semantic information from reports in the representation space*.
>
> We really agree with the reviewer on the viewpoint that discrepancy between 2D and 3D modalities could be much larger than that among classes. Therefore, Figure 6 in Appendix address this point and nicely demonstrate the effectiveness of UniMedI on integrating medical multi-modal images into a language-guided common semantic space, since Figure 6 shows semantic meaning is preserved while modalities are merged in the representation space, as we expected.
>
> > (**W2**) The details of the method are confusing (see Q2).
>
> > (**Q2**) The illustration of Figure 3 and Figure 4 is unclear to me. What are the light orange boxes used for? What is the operation after T_3D and T_2D? The main text in the method section is not corresponding to Figure 3 or Figure 4. It is unclear how Equations (1-2) were integrated into the framework.
>
> We appreciate the reviewer's suggestion to depict our method more clearly. Firstly, we would like to elaborate more on the pipeline of UniMedI to make the whole process clear. Based on the pipeline, we further explain contents in Figure 3 and Figure 4 clearly. After that, we address your question point-to-point in order to clarify any confusion. Lastly, **we have adjusted Figures 3 and 4 without changing the content in the updated version of our paper, including clearer annotations and some accompanying explanations**. We believe these enhancements will greatly improve the clarity and comprehensibility of our framework.
>
> ### Pipeline of UniMedI
> As we have described in the first paragraph of Section 3, to overcome the challenges that no paired 2D and 3D image data exists, during the training process, UniMedI first extracts a portion of informative 2D slices from a 3D volume according to language guidance via the proposed **attentive slice selection strategy (shown in Figure 4)**. This creates 2D-3D data pairs to bridge the disparity between the two modalities. Then, selected 2D slices and 2D X-rays, together with original 3D data, are sent to **a unified vision encoder (shown in Figure 3)** to obtain image representations.
>
> Overall, UniMedI is a vision-language pre-training framework. The paired report of each X-ray/CT is encoded by the text encoder, and VL contrastive learning (shown in Figure 3) are employed to obtain a language-guided common semantic space for both 2D and 3D medical images. To make this pipeline clear, we add a subfigure to Figure 3 for illustration.

---

> > ### Author Response · Authors · 2023-11-22
> > **Response to Reviewer zUjR**
> >
> > ### Figure 3 shows designs in the unified visual encoder and VL contrastive learning
> > For the vision encoder, first, 2D data (selected 2D slices and 2D X-rays) and 3D data (original 3D volumes) are processed by 2D tokenizer $T_{2D}$ and 3D tokenizer $T_{3D}$, respectively. Separate tokenizers encode input data from different modalities into a series of tokens. 3D CT tokens and corresponding 2D slices tokens are concatenated together for further processing by a shared backbone. 2D X-rays tokens directly processing by a shared backbone.
> >
> > Then, to enhance the cross-dimensional communication among 2D and 3D tokens, we adopt **an auxiliary task, i.e., mask and recovery**, and this task is implemented by the **self-distillation method**. Specifically, our visual encoder contains the teacher network $\overline{E}_v$ and the student network $E_v$, where $\overline{E}_v$ is updated by exponential moving averaged (EMA) over $E_v$. Tokens, once encoded by the tokenizer, are input into the teacher network. This process yields global and local embeddings and simultaneously outputs attention scores for each patch of the X-rays or CT scans, as outlined in **Equation 1** in main text. Depending on the scores of each patch, we mask the less significant areas (those with lower scores) for the input into the student network. Since **the student network only inputs important areas of the medical images**, the resulting visual encoding is more closely matched with the semantics included in the report. Finally, loss of the self-distillation method is applied to both the global $[CLS]$ token (i.e., $L_{icl}$) and local patch tokens (i.e., $L_{pcl}$).
> >
> > ### Figure 4 shows the attentive slice selection strategy
> > The aim of attentive slice selection is selecting informative 2D slices from the 3D CT volume. The 3D CT volume is processed by the tokenizer $T_{3D}$ and the teacher network, and thus, attention scores for patches in the CT can be obtained. Then, the scores of all tokens within one slice are averaged to derive the score of the corresponding slice. This process is referred to as the *Inter-Slice Average*, as outlined in **Equation 2** in main text. Slices with top $k$ score are selected as the connection between 2D and 3D data.
> >
> > Note that these scores are attention weights between the $[CLS]$ token and other patch tokens, and the $[CLS]$ token is used for VL contrastive learning. Therefore, this slice selection process is guided by text information in the report, and 2D slices containing important information in the report can be selected accordingly.
> >
> > >>2.a. What are the light orange boxes used for?
> >
> > In Figure 3, the orange boxes denote tokens of medical texts encoded by the language encoder $E_l$. The dark orange box represents the $[CLS]$ token, and the remaining light orange boxes represent all other tokens.
> >
> > In Figure 4, the orange box highlights a slice and the Inter-Slice Average process (explained in the above descriptions for Figure 4). In essence, the function of the orange box is to provide a clear understanding that this process occurs within a slice.
> >
> > >>2.b. What is the operation after T_3D and T_2D?
> >
> > Details are explained in the above descriptions for Figure 3. The merged line after $T_{2D}$ and $T_{3D}$ represents that 2D and 3D tokens are concatenated for further procesing by a shared backbone.
> >
> >
> > >>2.c. The main text in the method section is not corresponding to Figure 3 or Figure 4. It is unclear how Equations (1-2) were integrated into the framework.
> >
> > Equation 1 is utilized to produce attention scores for each patch in the teacher network. Equation 2 calculates the average attention score for patches in one slice, and this average score is used for slice selection. Refer to the above descriptions for Figure 3 and 4 for details.
> >
> > >(**W3**) Lack of baseline methods for 2D and 3D pre-training.
> >
> > >(**Q3**) As a paper for pre-training, the authors did not compare with the many existing pre-trained models, neither 3D or 2D. For 2D X-ray imaging, there are many pre-trained models publicly available [e.g., 1-3]. For 3D CT imaging, a variety of public models are also missing [e.g., 4-6].
> >
> >
> > Thanks for listing these works for us. We now provide comparisons and discussions to these work.

---

> ### Author Response · Authors · 2023-11-22
> **Response to Reviewer zUjR**
>
> Most listed models [1, 2, 4, 5] are trained via self-supervised learning (SSL). Comparing SSL models with the model by vision-language (VL) pre-training is not very reasonable, since VL pre-training is dealing with some multi-modality problem, which is quite different with the SSL problem. Moreover, UniMedI also handles multimodal medical images at the same time. Therefore, the comparison might not be as fair as what we listed in the paper. Nevertheless, we provide compared results in the below table (CheXpert$^\ast$ denoting full backbone fine-tuning and CheXpert$^\diamond$ representing linear classification, only fine-tuning the classification head). Here, we additionally include some latest SOTA SSL methods in the general domain, further demonstrating the superiority of our method.
>
>
> | Method | Perfomance | Description |
> | :--------: | :--------: | :--------: |
> | SimMIM [1]    | 88.07 (CheXpert$^\ast$, 100%) | Benchmark of SSL in Medical Image Classification |
> | SAM [2] | --- | Designed specific backbones for 2D and 3D data, only focus on landmark detection, Lesion matching |
> | Swin UNETR[4] | --- | Designed for specific task,  semantic segmentation of brain tumors in MRI Images |
> | DiRA [5] | 87.59 (CheXpert$^\ast$, 100%) | Unites discriminative, restorative, and adversarial learning in a unified manner |
> | Moco v2 [7]     |  84.90 (CheXpert$^\diamond$, 100%)    | General domain SSL method     |
> | Moco v3 [8]     |  82.02 (BCV, 100%) | General domain SSL method |
> | DINO [9]        |    82.61(BCV, 100%) | General domain SSL method |
> | UniMedI | **90.50** (CheXpert$^\diamond$, 100%), **85.40** (BCV, 100%) |---|
>
>
> It is evident that UniMedI outperforms all other methods, even without fine-tuning for the evaluation on CheXpert. Other methods not included in the above table are specially designed for some downstream tasks. SAM [2] is for landmark detection and Lesion matching with designed backbone, and it use separate models for 2D and 3D data, respectively. Swin UNETR [4] is designed for semantic segmentation of brain tumors in MRI Images, which cannot be applied to the same downstream tasks as our method. KAD [3] is designed for classification tasks, incorporating external knowledge and a classification head related to the disease. Med3D [6] is for transfer learning with a coarse-to-fine structure. Comparing with these specially designed methods is not a fair setting. Besides, all these designs for specific tasks are orthogonal with our methods, and can be involved in our framework for further performance improvement. Unifying all these designs can be our future work.
>
> >(**Q4**) The authors did not provide sufficient reference for the baseline methods presented in Table 1.
>
> Thanks for pointing out. We have added the references in the updated version.
>
> [1] Ma, DongAo, Mohammad Reza Hosseinzadeh Taher, Jiaxuan Pang, Nahid UI Islam, Fatemeh Haghighi, Michael B. Gotway, and Jianming Liang. "Benchmarking and boosting transformers for medical image classification." In MICCAI Workshop on Domain Adaptation and Representation Transfer, pp. 12-22. Cham: Springer Nature Switzerland, 2022.
>
> [2] Yan, Ke, Jinzheng Cai, Dakai Jin, Shun Miao, Dazhou Guo, Adam P. Harrison, Youbao Tang, Jing Xiao, Jingjing Lu, and Le Lu. "SAM: Self-supervised learning of pixel-wise anatomical embeddings in radiological images." IEEE Transactions on Medical Imaging 41, no. 10 (2022): 2658-2669.
>
> [3] Zhang, Xiaoman, Chaoyi Wu, Ya Zhang, Weidi Xie, and Yanfeng Wang. "Knowledge-enhanced visual-language pre-training on chest radiology images." Nature Communications 14, no. 1 (2023): 4542.
>
> [4] Tang, Yucheng, Dong Yang, Wenqi Li, Holger R. Roth, Bennett Landman, Daguang Xu, Vishwesh Nath, and Ali Hatamizadeh. "Self-supervised pre-training of swin transformers for 3d medical image analysis." In Proceedings of the IEEE/CVF Conference on Computer Vision and Pattern Recognition, pp. 20730-20740. 2022.
>
> [5] Haghighi, Fatemeh, Mohammad Reza Hosseinzadeh Taher, Michael B. Gotway, and Jianming Liang. "DiRA: Discriminative, restorative, and adversarial learning for self-supervised medical image analysis." In Proceedings of the IEEE/CVF Conference on Computer Vision and Pattern Recognition, pp. 20824-20834. 2022.
>
> [6] Chen, Sihong, Kai Ma, and Yefeng Zheng. "Med3d: Transfer learning for 3d medical image analysis." arXiv preprint arXiv:1904.00625 (2019).
>
> [7] Chen X, Fan H, Girshick R, et al. Improved baselines with momentum contrastive learning[J]. arXiv preprint arXiv:2003.04297, 2020.
>
> [8] Chen X, Xie S, He K. An empirical study of training self-supervised vision transformers. In 2021 IEEE[C]//CVF International Conference on Computer Vision (ICCV). 9620-9629.
>
> [9] Caron M, Touvron H, Misra I, et al. Emerging properties in self-supervised vision transformers[C]//Proceedings of the IEEE/CVF international conference on computer vision. 2021: 9650-9660.

---

### Public Comment · ~Che_Liu3 · 2023-11-22
**how does this work process bi-lingual data and using a 2D visual encoder for 3D image input?**

1. Different language of medical reports from 2D and 3D datasets:

The authors claim using BIMCV for 3D CT scan and MIMIC-CXR dataset for 2D CXR images, even they both have radiology reports, the reports in BIMCV is Spanish but MIMIC has English. The text encoder used in this work is BioClinicalBERT, pre-trained on English corpus, it is unable to tokenize the spanish text correctly as shown in Med-UniC(NeurIPS2023). Did authors translate all Spanish reports to English or keep it original?

2. How to utilizes 2D visual encoder for 3D CT input:

In the implementation section, authors utilize ViT-B/16 for extracting features from 2D images and 3D volumes. Does the 3D volumes is converted to several visual tokens for visual encoder input? If yes, the 3D visual tokens should be extremly long.

For example, the 2D images are tokenized to $16^{2}$ (256 in 1D) for each patch. To keep the 3D visaul token as the same shape in 1D as the 2D visual token, the 3D token might be $6^{3}$ and padded to 256 in total. However, the shape of the 3D volume is 128,128, 32, so the number of total 3D visual tokens could be over 2k. Does the GPU-V100 used in this work can handle a token squence over 2k under ViT for this pre-training? Because in 3D medical image analysis, normallly the batch size is quite small, like 4 in SwinUNTER, but in this work, it is 16(128 on 8GPUs). Is there any special operation to reduce the computational cost? The currect cost is too much to implement this model on V100.

---

> ### Author Response · Authors · 2023-11-23
> **Response**
>
> >(**Q1**) Different language of medical reports from 2D and 3D datasets:
>
> Thank you for your question. Indeed, you are correct. Given the language differences in the radiology reports from the BIMCV and MIMIC-CXR datasets, we have opted to translate all Spanish reports from the BIMCV dataset into English.
>
> >(**Q2**) How to utilizes 2D visual encoder for 3D CT input:
>
> Thank you for your valuable suggestion. We apologize if the description of our framework was unclear in the initial submission. For the vision encoder, first, 2D data (selected 2D slices and 2D X-rays) and 3D data (original 3D volumes) are processed by 2D tokenizer
>  and 3D tokenizer, respectively. Separate tokenizers encode input data from different modalities into a series of tokens. These tokens are then fed into the ViT-B architecture. Thanks to the presence of the 3D tokenizer, our CT scans are converted into 512 tokens, ensuring a manageable input size for the visual encoder.

---

> > ### Public Comment · ~Che_Liu3 · 2023-11-23
> >
> > Thanks for the detailed explaination. However, there are still some confused points.
> >
> > 1. Which model or software was used for translating Spanish medical reports to the English version? Based on our experiments, Google Translate is not designed for medical text translation, and the BIMCV reports have many Spanish medical abbreviations which are hardly translated. Since there are no machine translation models designed for this task, and no paired English-Spanish reports dataset with expert verification, how did the authors verify their translated medical reports? It would be helpful if the authors could visualize some translated samples or provide quantitative metrics on translated reports. Also, assistance from a radiologist who can speak both Spanish and English is necessary; otherwise, it is hard to determine if the translation is correct.
> >
> > 2. How did the model train with unpaired data?
> >
> > Thanks for the proposed attentive slice selection mechanism, the 3D CT scans can have paired 2D input and medical reports. However, other 2D images (approximately 217,000) from the MIMIC-CXR dataset do not have paired 3D CT scans. How did the model handle this situation during pre-training? It seems the number of 2D CXR images is much larger than that of paired 3D scans and 2D slices from CT. It would be beneficial for understanding this work if the authors could clarify the training procedure. Otherwise, it is difficult to understand the pre-training of unpaired 2D CXR images with CT scans.
> >
> > 3. Missing baselines from Reviewer zUjR
> >
> > In the comment from zUjr, it was stated that the comparison between the proposed method and SwinUNETR (CVPR 2022) is missing. SwinUNETR is a strong baseline for multiple 3D medical image segmentation tasks and is also implemented on the BCV dataset, which is used in this work. Why did the author state that SwinUNETR is specifically designed for MRI brain segmentation only? They have already provided their results on the BCV dataset. Additionally, nnUNet (Nature Methods) should be included as a standard baseline, as it is commonly used in most medical image segmentation tasks.

---

> > > ### Author Response · Authors · 2023-11-23
> > > **Response**
> > >
> > > >(**Q1**）Which model or software was used for translating Spanish medical reports to the English version? Based on our experiments, Google Translate is not designed for medical text translation, and the BIMCV reports have many Spanish medical abbreviations which are hardly translated. Since there are no machine translation models designed for this task, and no paired English-Spanish reports dataset with expert verification, how did the authors verify their translated medical reports? It would be helpful if the authors could visualize some translated samples or provide quantitative metrics on translated reports. Also, assistance from a radiologist who can speak both Spanish and English is necessary; otherwise, it is hard to determine if the translation is correct.
> > >
> > > We use the GPT4 to help to translate medical reports. Here are one case: "No signs of fractures or bone lesions were seen. On the chest CT scan, no signs suggestive of COVID-19 infection were observed. Isolated thickening of bronchioles with some isolated images of bronchorrhea were seen, findings of mild airway disease, bronchiolitis infection".
> > >
> > > >(**Q2**) How did the model train with unpaired data?
> > >
> > > In Figure 3, our model accepts 2D X-rays and pairs of 3D CT slices as inputs, and it doesn't require the X-rays and CT scans to be paired. Based on the alternate training approach of UniMiss, we have made improvements. Specifically, we mix 2D and 3D data within a single batch for training. This process does not require pairing between X-rays and CT scans. We learn the correlation between 2D and 3D through pairs of CT slices.
> > >
> > > >(**Q3**) Missing baselines from Reviewer zUjR.
> > >
> > > These two works are specially designed for segmentation tasks, while our work focuses on vision-language pre-training and handling multimodal medical images, which are quite different from these two works. It is unfair to compare with them on specific segmentation tasks which they are good at, and it is also unfair to compare with them on classification tasks which we are good at. UniMedI uses vision-language pre-training to offer the general ability to the model, rather than emphasizing some task-specific abilities during pre-training. Moreover, these designs for specific tasks are orthogonal with ours, and we might try to incorporate them in our future work.

---

> > > > ### Public Comment · ~Che_Liu3 · 2023-11-23
> > > >
> > > > Thanks for the clarification from authors.
> > > >
> > > > * Q1: Is the machine translated medical report accurate?
> > > >
> > > > Even though GPT-4 shows strong power in language translation tasks, the human evaluation procedure is still necessary and beneficial to understand the semantics. Otherwise, the model's ability is only reaching the translation upper limit, as mentioned in the review of an ICLR 2022 submission (https://openreview.net/forum?id=ZzwfldvDLpC). Although they used translated medical reports for other tasks, most reviewers questioned the quality of machine-translated reports and pointed out that human evaluation is necessary.
> > > >
> > > > * Q2: Confusion About Pre-training Strategy
> > > >
> > > >     - Thank you again for the explanation of the pre-training pipeline. The authors stated that the 3D CT scans are not aligned with any 2D CXR images, only with 2D CT slices. However, in the left of Figure 3, it appears that the model takes both 2D and 3D inputs at the same time and aligns their CLS embeddings. The authors claim that 2D and 3D data are mixed in a single batch. Is there a special mechanism to inform the model whether the input is 2D or 3D?
> > > >     - For instance, when the input is a 2D CXR, the model aligns the CXR with its paired CXR report.
> > > >     - When the input is a 3D CT scan, the model aligns the CT with its paired CT report and to select multiple 2D CT slices.
> > > >     - The model then aligns 2D CT slices with the 3D CT scan, but does not align the 2D CT slices with the paired CT reports.
> > > >
> > > > I am unsure if I have understood correctly because this pipeline seems like joint-training involving 2D, 3D, and text data together. In the batch, there are samples with pairs (2D CXR, Text) and triplet pairs (2D CT slices, 3D CT scan, Text). However, the BIMCV (3D dataset) only has over 8,000 samples with reports, whereas the MIMIC-CXR (2D dataset) has over 210,000 samples with reports. The severe imbalance could affect the pre-training, as most batches would only have very few 3D samples. This problem is not clarified in this work, and it is really important for others who want to follow this research.
> > > >
> > > > * Q3: Missing baselines in 3D tasks
> > > >
> > > >     - The authors claim that SwinUNTER is not specifically designed for 3D CT classification tasks. However, in Alice's work (ICCV 2023 oral), SwinUNTER has already been implemented in 3D CT classification, achieving substantial improvement compared to MocoV3 (referenced in Table 9, appendix of Alice's work), which serves as a baseline in this study. Furthermore, SwinUNTER is designed to pre-train a visual encoder for various downstream tasks, meaning it can be directly transferred to classification tasks. It is not specifically intended for segmentation tasks, as its pre-trained visual decoder is discarded in downstream segmentation tasks.
> > > >
> > > >     - Also, PCRLv2 (TPAMI 2023) has implemented the LUNA16 classification task, similar to the evaluation method used in this work for assessing pre-trained model performance. Therefore, how can UniMedI, pre-trained with both 3D CT scans and 2D slices along with paired radiology reports, demonstrate superiority over other baselines that only perform self-supervised learning (SSL) on 3D CT scans? Given that the 2D slices are derived from 3D CT scans, it is challenging to assert the introduction of new information in the pre-training process. Moreover, such baselines as SwinUNTER and PCRLv2 have already shown strong performance in 3D segmentation and classification tasks using image-only SSL. This work might require more contemporary baselines to showcase its superiority in 3D tasks through the proposed pre-training strategy.
> > > >
> > > > * Q4: Missing baselines in 2D tasks
> > > >
> > > >     - For the 2D baselines, MRM (ICLR 2023), MedKLIP (ICCV 2023), and PRIOR (ICCV 2023), they all show strong performance on RSNA segmentation and CXR image classification tasks, which have been used to evaluate the pre-trained model in this work. Why are they missing?

---

### Author Response · Authors · 2023-11-23
**General Response**

We thank all the reviewers for their time and thoughtful feedback. We revised our manuscript and submitted a new version for review. We provide point-to-point responses to each review in detail. We revised our manuscript and submitted a new version for review. We highlight the changes in blue color.

# Summary of Revision.
## Main Paper

1. We highlight the difference among three t-SNE figures. (Reviewer zUjR)
1. We adjust Figure 3 and 4 without changing the content and add a subfigure in Figure 3, including clearer annotations and some accompanying explanations. (Reviewer zUjR, EXCm, 8U1f, vK4s)
1. We polish the writing in our main paper. (Reviewer EXCm)
1. We provide sufficient reference for the baseline methods presented in Table 1. (Reviewer zUjR)
1. We state the version of RSNA in EXPERIMENTAL SETUP. (Reviewer EXCm)
1. We fix some typos.

## Appendix

1. Appendix B: We add more implementation details of downstream tasks, including detailed training setting and segmentation decoder. (Reviewer vK4s)
2. Appendix C.1: We provide pneumonia detection in RSNA. (Reviewer EXCm, vK4s)
3. Appendix C.2: We give experimental result in larger dataset, AMOS. （Reviewer EXCm, 8U1f, vK4s)
4. Appendix C.3: We provide different metrics in COVIDx. (Reviewer 8U1f)
5. Appendix C.4: We visualize data representations of two different classes (i.e., pneumonia and cardiomegaly). (Reviewer zUjR, 8U1f)

---

### Meta-Review · Area_Chair_kL81 · 2023-12-10

**Metareview:**

The paper presents a novel framework, UniMedI, for pre-training that unifies 2D (X-ray) and 3D (CT) medical imaging modalities using language embeddings. This approach aligns features of unpaired images with the same pathological conditions into a similar space, leveraging associated diagnostic reports. The results demonstrate improved performance in transfer learning tasks across both 2D and 3D medical imaging.

The framework's ability to handle diverse imaging modalities and leverage rich semantic information from diagnostic reports is a significant contribution to the field of medical imaging. The framework's performance is evaluated across multiple datasets, showing its effectiveness in various medical imaging tasks.

The novelty of the approach is questioned, with some reviewers noting that it may be incremental or similar to existing methods like Attentive Mask CLIP. The paper does not sufficiently compare its approach with existing pre-trained models in 2D and 3D medical imaging, which is a significant oversight for a paper focusing on pre-training methods. Some reviewers raised concerns about the practical significance of the improvements shown, suggesting that the results indicate only marginal enhancements.

The authors have addressed several concerns raised by the reviewers, particularly around the clarity of the method and the distinction from existing methods. However, some reviewers remain unconvinced about the novelty and the practical significance of the improvements offered by the framework.

The paper introduces an interesting and clinically relevant approach to medical image analysis, addressing the challenge of unifying 2D and 3D medical imaging modalities. While the idea is promising and the paper is generally well-executed, concerns about the clarity of the method, its novelty, and the significance of its contributions remain. Given the mixed reviews and the authors' responses, the decision on this paper should be carefully considered, weighing its potential impact against the concerns raised.

**Justification For Why Not Higher Score:**

The novelty of the approach is questioned, with some reviewers noting that it may be incremental or similar to existing methods like Attentive Mask CLIP. The paper does not sufficiently compare its approach with existing pre-trained models in 2D and 3D medical imaging, which is a significant oversight for a paper focusing on pre-training methods. Some reviewers raised concerns about the practical significance of the improvements shown, suggesting that the results indicate only marginal enhancements.

**Justification For Why Not Lower Score:**

N/A

---

### Decision · Program_Chairs · 2024-01-16

Reject